

# The Role of Topography, Land and Sea Surface Temperature on Quasi-Stationary Waves in Northern Hemisphere Winter: Insights from CAM6 Simulations

Cuiyi Fei[1] and Rachel H. White[1]

[1]Department of Earth, Ocean and Atmospheric Sciences, University of British Columbia

**Correspondence:** Cuiyi Fei (cfei@eoas.ubc.ca)

**Abstract.** Quasi-stationary waves (QSWs), atmospheric Rossby waves with near constant phase that persist on subseasonal timescales, are not distributed homogeneously across the globe, even at a given latitude. The climatological QSW amplitude has a distinct spatial pattern, with clear zonal asymmetries, particularly in the Northern Hemisphere; those asymmetries must be impacted by stationary forcings such as land, topography, and sea surface temperatures (SSTs). To investigate the effects of

stationary forcings on QSW characteristics, including their duration and spatial distribution, eight simulations were conducted using CAM6 with prescribed SSTs. These simulations range from realistic, semi-realistic (with some stationary forcings matching reality) to fully idealized (with idealized forcings added in aquaplanet simulations). The control simulation was validated against ERA5 reanalysis data. Stationary forcings tend to extend the duration of QSWs and move the zonal-mean amplitude of QSW northward in the midlatitudes. The stationary forcings also strongly impact the zonal asymmetric distribution of QSWs.

QSWs are primarily influenced by both the local stationary wavenumber $Ks$, which depends on jet speed and its second-order meridional gradient, and by the strength of transient eddies. In some cases, QSW strength is also associated with the strength of the stationary waves. When the timescale of the QSWs is changed (e.g. from 15-30 days to >30 days), the relative contributions from different mechanisms changes, but stationary wavenumber $Ks$ and transient eddies strength are important in all time scales for experiments with realistic land.

## 1  Introduction

Quasi-stationary waves (QSWs) are extra-tropical Rossby waves with near-constant phase on time scales of several days to weeks and can be closely associated with the occurrence of extreme temperature, precipitation, and wind events (Petoukhov et al., 2013; Kornhuber et al., 2017; Wolf et al., 2018; White et al., 2022; Li et al., 2024). Despite their importance as potential precursors to extreme weather events, the mechanisms driving QSWs remain not fully understood. Previous research on the

features and mechanisms of atmospheric circulation patterns with similar time scales may help inform our understanding of QSWs, including atmospheric blocking events and planetary scale wave trains or teleconnection patterns (Ali et al., 2022; Teng et al., 2013; Pfahl, 2014; Röthlisberger et al., 2019). In contrast to atmospheric blocking, QSWs have a more longitudinally extended wave-like structure, typically with several consecutive high and low pressure anomalies. The repeated occurrence of QSWs with similar phases over the same region can result in seasonal-averaged patterns, which can be identified as telecon-



nection patterns (Xu et al., 2019). It is also clear that many features of QSWs, e.g. their frequency, are not uniform across longitudes (Wolf et al., 2018; Fei and White, 2023), as is seen for atmospheric blockings and teleconnections. This suggests a clear role for stationary forcings in shaping the frequency and/or strength of QSWs, including topography (Luo and Chen, 2006; Narinesingh et al., 2020) and diabatic heating (O'Reilly et al., 2016).

Given the similarities between QSWs and atmospheric blockings and/or teleconnection patterns, it is natural to hypothesize
that aspects of the climate that influence the spatial distribution of atmospheric blocking and/or teleconnections could also influence QSWs. From the perspective of the internal dynamics of low-frequency variability of the atmosphere, which broadly covers 10-90 days time scales (although the exact timescale varies in different research), barotropic instability, non-modal instability growth, and background flow conditions can all contribute to blockings and/or related subseasonal variability (Simmons et al., 1983; Branstator, 1990, 1992; Swanson, 2000, 2002; Nakamura and Huang, 2018). The nonlinear activity of transient
eddies interacting with the slowly-varying background flow can play a critical role in the formation of atmospheric blocking directly, particularly in the North Pacific (Hwang et al., 2020). Nakamura and Huang (2018) developed a theory describing the lifecycle of atmospheric blocking and providing a framework to quantify the conditions for blocking formation, considering both background flow and Rossby wave activity. Thus it seems that both background flow conditions, and the presence of instability and/or transient eddies are likely important for the presence of atmopsheric blocking and/or teleconnections, and
thus, perhaps for QSWs. The zonal distribution of these background flows, instabilities and transient eddies is determined by zonal asymmetries, which are ultimately shaped by stationary forcings (Brayshaw et al., 2009, 2011), including topography and surface temperature gradients.

Spatial patterns in surface temperature and related precipitation, particularly convective precipitation, lead to spatial patterns in atmospheric diabatic heating, which is composed of radiation, sensible heating and latent heating. Linear theory explains how
diabatic heating in the tropical upper troposphere, primarily stemming from latent heat release during tropical convection, can generate Rossby waves. These waves can propagate away from their source region and may become trapped within midlatitude waveguides (Hoskins and Karoly, 1981; Sardeshmukh and Hoskins, 1988; Hoskins and Ambrizzi, 1993; Jin and Hoskins, 1995; Ding and Wang, 2005; Branstator, 2014). A key example is the Pacific-North America (PNA) pattern, an atmospheric low-frequency mode of variability consisting of a wavetrain extending from the tropical Pacific across North America, which
can be driven by diabatic heating sources in the tropics (Franzke et al., 2011). These heating sources may be influenced by tropical internal variability, such as El Niño-Southern Oscillation (ENSO), linking them to sea surface temperature (SST) patterns (Hoskins and Karoly, 1981; Wallace and Gutzler, 1981; Franzke et al., 2011; Wolf et al., 2022). Interestingly, the time scale of the midlatitude Rossby wave response to tropical heating can be much longer than the time scale of the tropical heating (Branstator, 2014). Additionally, anomalous SST patterns in the mid-latitudes can serve as heating sources, driving
downstream Rossby wave trains, where the roles of both SST patterns and land-ocean contrast are highlighted (Xu et al., 2019). Anomalous soil moisture can also create anomalous diabatic heating and drive the circumglobal teleconnection (Teng and Branstator, 2019)

In addition to diabatic heating, the influence of topography on Rossby waves at subseasonal time scales, particularly related to blockings, has been a focus of research since the last century (e.g. Charney and DeVore, 1979; Charney and Straus, 1980).



Charney and DeVore (1979) have shown that topography can induce a quasi-equilibrium state with a relatively weaker zonal component locked close to linear resonance, which is believed to represent atmospheric blocking in this highly idealized studies. In more realistic studies, topography has been linked to the phase-locking behavior of QSWs and the weakening of westerlies, which can result in localized summer heatwaves (Jiménez-Esteve et al., 2022). Luo and Chen (2006) suggests that topography helps anchor the locations of atmospheric blocking; however, it has also been found in one model that, while

topography may increase the frequency of atmospheric blocking events, it does not necessarily fix their specific locations (Narinesingh et al., 2020). There is relatively little research about the role of topography on teleconnections (Jin and Ghil, 1990).

    In addition to the influence on subseasonal variability, potentially including QSWs, as described above, topography and asymmetries in diabatic heating (including land-sea contrast and gradients in SSTs) are the primary drivers of Earth's stationary

waves (Garfinkel et al., 2020; White et al., 2021). According to quasi-resonant amplification (QRA) theory, under particular background conditions, stationary waves may be able to resonant with transient eddies to produce high amplitude QSWs (Petoukhov et al., 2013, 2016). Thus topography and diabatic heating sources may have additional influences on QSWs, through their impact on the stationary waves.

    The impact of stationary forcings on blockings or subseasonal variabilities, and thus likely on QSWs, does not imply that

QSWs cannot exist without such forcings. Indeed, Held (1983) and Wolf et al. (2022) both demonstrate the existence of QSWs in aquaplanet climate model simulations. From a simple perspective of Rossby waves' phase and group speeds, Rossby waves can become (quasi-) stationary if the intrinsic westward propagation of the Rossby waves is countered sufficiently precisely by the eastward advection of the wave by the background zonal winds, resulting in a wave with near-zero phase speed (Hoskins and Ambrizzi, 1993; Hoskins and Woollings, 2015). However, as highlighted by Röthlisberger et al. (2019), the phase speed

and group velocity of Rossby waves cannot both be zero simultaneously. This suggests that when Rossby waves become quasi-stationary, i.e., when the phase speed is zero, their group velocity should be positive, resulting in downstream transport of the wave energy. This therefore necessitates a persistent energy source to sustain the QSWs. This implies that the presence of QSWs can be the result of multiple factors: a wave source (e.g. tropical convection); suitable propagation conditions (zonal wind); and perhaps also suitable conditions for wave growth (e.g. meridional SST gradients); positive feedbacks between these

factors may also be key (Zappa et al., 2011).

    While different theories exist regarding how stationary forcings influence blockings and other subseasonal variabilities, their significant impact is widely recognized. Building on the literature of the roles of diabatic heating and topography in shaping blockings and various subseasonal phenomenon, we can hypothesize that stationary forcings may provide a crucial foundation for understanding the mechanisms behind QSWs. To investigate this, we conduct eight climate model experiments

with different stationary forcings; these experiments are detailed in the Data and Methods section. In the Results section, we present the fundamental features of QSWs observed in these experiments, compare their characteristics, and understand how stationary forcings shape the key QSW features, including zonal mean amplitude and the spatial distribution. We focus on the Northern Hemisphere (NH) as the stationary forcings are stronger than in the Southern Hemisphere, and on winter, as QSWs in the NH have been shown to have the largest amplitude in winter (Wolf et al., 2022).





**Table 1.** CAM6 simulations

| Experiment name | prescribed SST | topography | land |
|---|---|---|---|
| CTRL | control SST | real-world | real-world |
| ZSST | zonally symmetric control SST | real-world | real-world |
| NM | control SST | no topography | real-world |
| NM-ZSST | zonally symmetric control SST | no topography | real-world |
| RWM | zonally symmetric control SST+lapse rate | real-world topography | no land |
| IWM | zonally symmetric control SST+lapse rate | idealized mountain (180-150W) | no land |
| HEAT | High and low SST centres | no topography | no land |
| AQUA | zonally symmetric control SST | no topography | no land |

## 2 Data and methods

### 2.1 Idealized experiments

In this research, we conducted eight 100-year simulations using the Community Atmosphere Model version 6 (CAM6) (Bogenschutz et al., 2018) with prescribed SSTs, with the first 10 years discarded as spin-up period. All experiments with realistic land-ocean distribution used the CESM2 compset F2000climo (atmospheric climatological conditions for the year 2000 with CAM6 physics and finite-volume dynamics core), while aquaplanet simulations used the compset QPC6 (prescribed SST aquaplanet using CAM6), all at a resolution of 1.9x2.5 degrees. While this resolution is relatively coarse, it is sufficient to capture the large-scale circulation patterns, including the QSWs we focus on in this study. ERA5 reanalysis data (Hersbach et al., 2020) is used to compare with a control (CTRL) experiment to confirm that the model captures the main features of observed QSWs. The details of the eight experiments are summarized in Table 1.

In the CTRL simulation, the full atmosphere and land models are coupled with climatologically prescribed SSTs, including a seasonal cycle. The prescribed SSTs used in the compset F2000climo represents averaged SST with a seasonal cycle from the 1995-2005 Hadley Centre Sea Ice and Sea Surface Temperature dataset (HadISST). We perform semi-realistic experiments to isolate the influence of different stationary forcings: in the zonal-SSTs (ZSST) experiment, zonal SST asymmetries are removed; in the no-mountain (NM) experiment, all topography is removed, including the effects of sub-grid scale topography; and in the NM-ZSST experiment, both topography and zonal SST asymmetry are removed. All zonally symmetric SSTs used in our experiments are zonal averages of the observed SSTs, ensuring comparability with the realistic SST experiments. An aquaplanet (AQUA) experiment was also conducted to investigate QSWs without any stationary forcing.

To investigate the importance of land-ocean contrasts and understand stationary forcings in highly idealized situations, we also conducted two "water mountain" experiments based on AQUA. In these experiments, topography was imposed, but without land; a lapse rate of 6.5 K per kilometer was added to the SSTs to avoid anomalous heating sources in the troposphere on top of mountains. Variations of the water mountain experiments—RWM (realistic water mountain) and IWM (idealized water mountain)—were designed to explore the potential impact of topography's shape. The RWM experiment incorporates





water mountains that mimic realistic topography. The IWM experiment features a Gaussian mountain spanning 30°N–60°N and 180°W–150°W, with a rectangular plateau at 2 km height in the middle, offering an idealized configuration. In the two idealized mountain experiments, the surface geopotential ("PHIS" in CAM6) was calculated based on the AQUA simulation, and only modified in the region of the corresponding added topography. To isolate the impacts of SST anomalies, the HEAT experiment imposed a 20 K positive SST anomaly at 30°N–60°N, 180°W–150°W, paired with a 20 K negative SST anomaly at 30°N–60°N, 150°W–120°W on the aquaplanet. This setup replicates the strongest NH winter surface temperature contrast (Siberia vs. Pacific); while the real contrast is weaker, especially at lower latitudes, we use this idealization for a clearer response.

## 2.2 Quasi-stationary waves, stationary waves and transient eddies

We calculate quasi-stationary waves strength by assessing the amplitude of persistent Rossby wave packets, using the method outlined by Zimin et al. (2003). The quasi-stationary Rossby wave packet amplitude is calculated from the 15-day Lanczos lowpass filter (Duchon, 1979) applied to the day-of-year anomalous meridional wind at 200 hPa, with only wavenumbers 4-15 included, following Röthlisberger et al. (2019). By using the 200hPa pressure level, we are studying QSWs slightly higher in the atmosphere than Wolf et al. (2018), who studied QSWs at 300hPa; we compare the results at different levels and wavenumbers in the Summary and Discussion. The QSW strength is calculated as:

$$|2\sum_{k=4}^{15} v_{tf}(k,t)e^{2\pi ikl_\lambda/N}| \tag{1}$$

where $t$ denotes time, $v$ is meridional wind and $v_{tf}$ denotes the 15-day lowpass filtered meridional wind. $k$ denotes wavenumber and $l_\lambda$ denotes the longitudinal grid point index varying from 0 to $N$, where $N$ is the number of longitudinal grid points.

To study the duration of QSWs, we define QSW events following the methodology outlined in our previous work (Fei and White, 2023), with one modification: the latitude range is adjusted from 35°–65°N to 30°–60°N. This change accounts for the distribution of QSWs in winter in most experiments, which is more concentrated within 30°–60°N. In contrast, Röthlisberger et al. (2019) chose 35°–65°N because their study focused on summer in realistic world. Quasi-stationary wave events are identified as a series of $n$ consecutive days during which the averaged QSWs amplitude within a region covering 30° of latitude and 60° of longitude exceeds a threshold $x$. To ensure robustness, we explored two approaches of threshold: fixing either the event frequency or the amplitude threshold $x$. When the event frequency was fixed at an average of one event per year (90 events in a 90-year run), the amplitude threshold $x$ varied across different experiments. Conversely, when the amplitude threshold was fixed, the number of events increased with stronger stationary forcings, leading to the selection of more and/or longer events.

To maintain a consistent measurement of wave strength, the strength of wavenumbers 4-15 stationary waves and transient eddies were also calculated using Zimin's method, with the 15-day lowpass filtered anomalous meridional wind replaced by the day-of-year climatology for stationary waves and the 15-day Lanczos highpass filtered meridional wind for transient eddies.



## 2.3 Stationary wavenumber and Eady growth rate

In a barotropic framework, the stationary wavenumber $Ks$, calculated from the background flow conditions, describes the maximum zonal wavenumber of a Rossby wave that can become stationary (Hoskins and Ambrizzi, 1993), i.e. $Ks^2 = k^2 + l^2$ for a stationary wave, where $k$ and $l$ are the zonal and meridional wavenumbers respectively. We analyse the impact of changing stationary wavenumbers on the distribution of QSWs in our different experiments. We calculated the climatology of stationary wavenumber $Ks$ across all experiments using daily zonal wind data filtered with a 15-day low-pass filter to reduce the non-linear impacts of high frequency waves, based on the following definition:

$$Ks = a\sqrt{\frac{\beta_m}{U_m}}, \beta_m = \frac{2\Omega cos^2\phi}{a} - \frac{cos\phi}{a^2}\frac{\partial}{\partial\phi}\frac{1}{cos\phi}\frac{\partial}{\partial\phi}(U_m cos^2\phi)$$ (2)

where $a$ denotes the radius of the Earth, $\phi$ denotes the latitude, $\beta_m$ is the meridional gradient of absolute vorticity in Mercator projection, and $U_m$ is the zonal wind in Mercator projection.

To study the potential for the growth of waves, we use the Eady growth rate, derived from the Eady Model (Eady, 1949), which effectively represents local baroclinic instability. It can be defined using zonal wind or temperature by an approximation form of thermal wind balance and we further use hydrostatic balance and ideal gas law to approximate the Brunt-Väisälä frequency $N$. Here, we adopt the following definition (Lindzen and Farrell, 1980):

$$\sigma_E = -0.31 * \frac{g}{NT}\frac{\partial T}{\partial y}, N = \sqrt{\frac{g}{\theta}\frac{\partial\theta}{\partial z}} = \sqrt{-\frac{g^2 p}{RT\theta}\frac{\partial\theta}{\partial p}}$$ (3)

where $\sigma_E$ denotes Eady growth rate, $g$ denotes the acceleration due to gravity, N is the Brunt-Väisälä frequency, $T$ represents the temperature, $y$ is the northward distance, $z$ represents the vertical coordinate, $p$ represent the pressure coordinate, and $R$ is the gas constant for dry air. We choose the Eady growth rate at 700 hPa in our analysis because it represents a low level where baroclinic instability is strong in most situations (e.g., Simmonds and Lim, 2009).

## 3 Results

### 3.1 The duration of quasi-stationary wave events

We first explore the duration and frequency of discrete QSW events in the different experiments. We focus on the North America region (Fig. 1), but conclusions for a region over Europe (see Fig. A1) are very similar. When the frequency of all events is fixed (first row in Fig 1), the CTRL CAM simulations tends to over-simulate the frequency of short events relative to ERA5, and under-simulate the frequency of long events. This is increasingly notable as the minimum duration threshold is decreased (Fig. 1a to 1c). In the second row, where events are selected based on a fixed amplitude, this bias remains, with ERA5 typically showing lower frequency than the CTRL simulation for shorter events, and higher frequency for longer events. This bias, of too many short events, and too few long events, can also be seen in the 3rd row, where the black dashed line shows





the ERA5-CTRL difference when using a fixed amplitude threshold, highlighting a potential bias in CESM2's representation of QSW persistence. This bias is consistent when selecting different regions (see, e.g., Fig. A1).

Figure 1g, 1h, and 1i illustrate the differences in QSW duration distributions with fixed amplitude between the CTRL experiment and other experiments (as well as ERA5 minus CTRL results) to show the role of stationary forcings on QSW
durations more clearly. Regardless of the applied duration threshold (different plots), adding stationary forcings consistently increases the frequency of QSWs with relatively longer durations, peaking around 12 days, while slightly decreasing the frequency of QSWs with shorter durations (Figs. 1i). Specifically, in AQUA the frequency of QSWs with an 8-day duration threshold is lower than in any other experiment, whereas with a 1-day duration threshold, the frequency of QSWs in AQUA is higher than in any other experiment. Note that here, duration refers to the QSW amplitude exceeding a threshold for a set
number of days; the 15 day low pass filter in the QSW definition still requires persistence on this timescale, even for a 1 day event. The disproportionate increase and decrease in QSW events illustrates an overall increase in the absolute number of events as well when adding stationary forcings. This highlights the critical role of stationary forcings in either generating more frequent or stronger QSWs (as evident in Fig. 1g, where the frequency of QSWs almost purely increases) or sustaining QSWs for longer durations (indicated by the reduction in frequency of events shorter than 10 days and increase for those exceeding 10
days in Figs. 1h and 1i), or both. Alternative regional definitions yield similar results, indicating the robustness of the analysis. The repeated analysis for Europe (0–60°E) is shown in Figure A1, and the main conclusion remains robust: stationary forcings play a key role in generating more frequent or stronger QSW events and/or sustaining their duration.

## 3.2  The zonal amplitude of quasi-stationary waves

Since the added topography and land surfaces vary with both latitude and longitude, we now investigate whether these forcings
can influence zonal mean features, before progressing to understanding their impact on the zonal asymmetries of QSWs. Figure 2 displays the zonally averaged QSW amplitude, along with the climatological zonal wind and meridional wind amplitude all as a function of latitude. We analyse the zonal wind and meridional wind amplitude, as one way stationary forcings can influence QSWs is likely through their impact on atmospheric circulation. We study absolute meridional wind amplitude as a proxy for stationary wave amplitude. Overall, the CTRL simulation (red dashed line) simulates reasonably well the zonal mean QSW
strength relative to ERA5 (black dashed line), albeit with a slight underestimation in the extratropics, peaking around 60N (see ERA5-CTRL differences in the black dashed line in Fig. 2d). Zonal mean zonal winds are also well simulated (Fig. 2b and e) with differences smaller than 10%. The stationary wave strength is underestimated in much of the extratropics (Fig. 2c and f), particularly around 60N, where the zonal winds were overestimated and QSWs are also underestimated. These results do not necessarily indicate that extratropical QSWs are underestimated at all longitudes, and the full spatial distribution will be
discussed in the next subsection.

Interestingly, there are relatively small differences in the zonal mean QSW strength between different experiments (Fig. 2a and d), although stationary forcings typically increase QSW amplitude in the extratropics (Fig. 2d), consistent with the results in Section 3.1. There are more notable differences in zonal mean zonal wind (Fig. 2b and e) and meridional wind amplitude (Fig. 2c and f). In Figure 2, red lines indicate simulations that have the realistic land and ocean distribution, whilst blue lines





**Figure 1.** The duration-frequency distribution of QSW events across different experiments within the "North America" region, defined as the area between 60°W and 120°W.The first and second rows show distributions for thresholds of (a)/(d) 8 days, (b)/(e) 4 days, and (c)/(f) 1 day on the QSW metric. In the first row, events are selected based on a fixed frequency, while in the second row, events are selected using a fixed amplitude threshold. The third row shows ERA5-CTRL, as well as the difference between the CTRL and other experiments (as CTRL - experiment) for thresholds of (g) 8 days, (h) 4 days, and (i) 1 day using the fixed amplitude threshold. The lines are smoothed with 3-day bins on the x-axis. The color coding is as follows: ERA5 is shown in black, experiments with realistic land-ocean contrast are in red, experiments without land surfaces in blue. Dashed lines represent experiments with zonally asymmetric SSTs, while solid lines indicate those with zonally symmetric SSTs. Thick lines correspond to experiments that include topography, whereas thin lines represent experiments without topography.

indicate different aquaplanet simulations. Thus, comparisons between the red lines and blue lines at the first row (with the





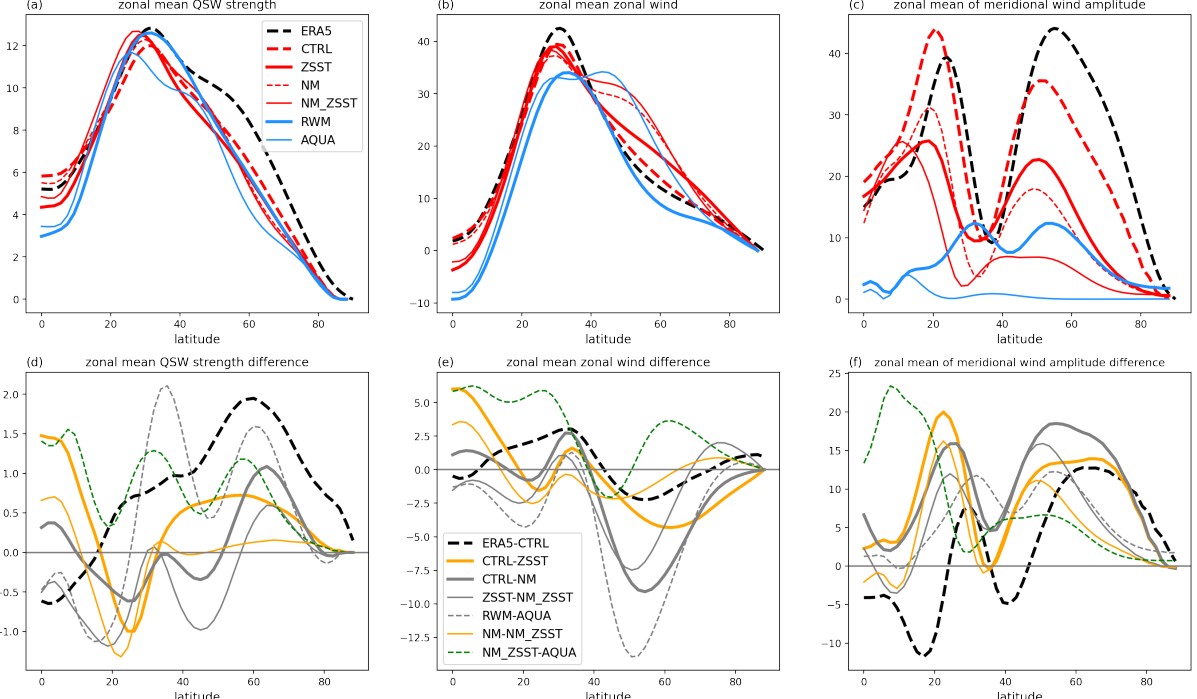

**Figure 2.** The top row shows the zonal mean of (a) QSW strength, (b) zonal wind, and (c) meridional wind amplitude. The bottom row shows the difference between experiment pairs and ERA5 minus CTRL, with zonal mean values in ERA5 interpolated to the spatial resolution of the CAM6 simulations. The line colors and styles are the same as in Fig. 1; in the second row, gray represents experiment pairs highlighting the impact of topography (CTRL–NM, ZSST–NM_ZSST, RWM–AQUA), orange represents experiment pairs showing the impact of zonal SST asymmetry (CTRL–ZSST, NM–NM_ZSST), and green (NM_ZSST–AQUA) represents the impact of land surface.

same line style) illustrates the impact of the land surface (also given by the green line in the second row) - a weakening of the jet streams around 40°N, and strengthens the stationary waves and QSWs at all latitudes. The difference between the thick and thin lines at the first row, or analysing the gray lines in the second row, illustrates the impact of realistic topography: a deceleration of the zonal wind around 50–60°N and subsequent changes in the meridional gradient of zonal wind. Topography

also increases the stationary wave amplitude, as expected (see Figs. 2c and 2f). The impact on QSWs is very latitude dependent, but with a consistent increase around 60N across all experiment pairs (Fig. 2d). The effect of zonal SST asymmetry, illustrated by comparing solid lines with dashed lines at the first row, or analysing the orange lines in the second row, is relatively small for QSWs and zonal mean zonal wind, but more substantial for the meridional wind amplitude. This is likely because zonal SST asymmetry (illustrated by comparing solid lines with dashed lines) influences jet streams by adjusting the regional meridional

SST gradients without changing the zonal mean meridional SST temperature gradient, whereas topography and land surface act as additional zonal forcings directly.





Figure 2d and 2e show some indication that, when looking at the differences between experiments, in the extratropics, latitudes where the zonal wind is weaker in CTRL correspond to latitudes where QSW strength is stronger, and vice versa. This suggests a potential link between QSW strength and zonal wind speed. Additionally, contrasting Figures 2c and 2f, poleward of 30°N, larger increases in QSW strength typically occur with larger increases in meridional wind amplitude, supporting an association between QSWs and stationary wave strength. These results thus suggest that QSW strength may be associated with zonal wind speed and stationary wave strength. On the other hand, the CTRL simulation bias in zonal and meridional winds is not proportional to the bias in QSWs strength, suggesting a complex interaction between circulation and QSWs strength.

### 3.3 The spatial distribution of quasi-stationary waves

After examining the zonal mean amplitude of QSWs and noting relatively small differences between most experiments, we now focus on their spatial distribution and the effects of stationary forcings on this distribution. Based on previous theories of teleconnections and blocking, we test two hypotheses connected to the zonal and meridional wind:

1. The spatial distribution of QSWs is associated with the spatial distribution of stationary waves, due to a similar influence of stationary forcings on both stationary and quasi-stationary waves, e.g. through zonal wind changes, or due to wave resonance between stationary waves and transient eddies resulting in QSWs.

2. The spatial distribution of quasi-stationary waves is governed by a combination of stationary wavenumber ($Ks$) and transient eddy strength/local Eady growth rate, as Rossby waves with given wavenumbers can become stationary under background conditions with the corresponding stationary wavenumbers.

To test these hypotheses, four variables—stationary wave strength, stationary wavenumber ($Ks$), transient eddy strength, and Eady growth rate—are analyzed. We analyse 'stationary wave strength' to assess the first hypothesis. Conversely, analysis of correlations between QSWs and the 'stationary wavenumber $Ks$' and 'transient eddies strength' relate to the second hypothesis, along with 'Eady growth rate', as high values of Eady growth rate indicate regions where Rossby waves may grow rapidly. The potential of Rossby wave growth may be important as Hypothesis 2 applies regardless of whether strong Rossby waves propagate into the region (as transient eddies) or are generated locally. Thus, Eady growth rate helps assess whether local instability contributes to explaining the spatial distribution of QSWs.

Before analyzing the differences across experiment groups, we first evaluate whether our CTRL simulation effectively replicates the observed spatial distribution of QSWs, and the four metrics of interest. Overall, the zonal and meridional wind climatologies are well-represented in CTRL (compare Figs. 3a and 3b). The QSWs strength (shown in the coloured shading in Figs. 3c-j), is also generally well represented, although with an under-simulation of climatological QSW strength over Europe and the Atlantic; this is somewhat answered in the Summary and Discussion when comparing QSWs with different wavenumbers at different pressure levels. For stationary wavenumber $Ks$ (black contours in Figs. 3c and 3d), the regions where $4 <= Ks$ $<= 8$ are again generally well represented, although the CTRL simulation underestimates the value of $Ks$ at many high-latitude regions. We focus on regions where $4 <= Ks <= 8$ due to the high correspondence of this range with regions of strong climatological QSW strength, as can be seen in both ERA5 (Fig. 3c) and CTRL (Fig. 3d). In Figs. 3e and 3f, we see that the transient





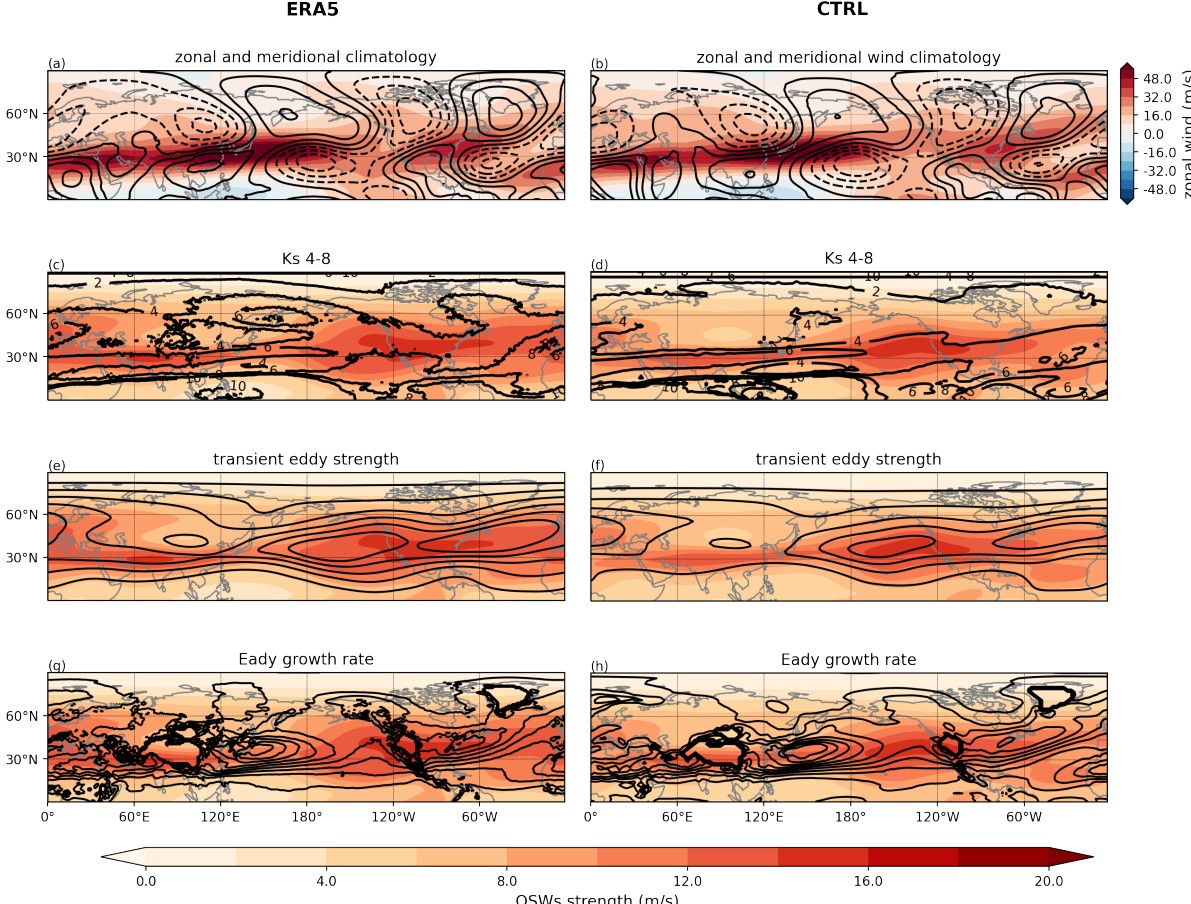

**Figure 3.** Climatologies for DJF in ERA5 (left column) and the CTRL simulation (right column). Panels (a) and (b) depict the zonal wind climatology (shading) and meridional wind climatology (contours) with different colorbar. In subsequent plots, shading consistently represents QSW strength. Contours in (c) and (d) show the stationary wavenumber $Ks$ range of 4–8, (e) and (f) show transient eddy strength, (g) and (h) display the Eady growth rate at 700 hPa, and (i) and (j) illustrate the zonal gradient of zonal wind. The left column presents features from ERA5, while the right column corresponds to CTRL. The Eady growth rate is masked over regions where topography exceeds 1500 m. The contour interval in (a)(b) is 3 m/s, with negative values dashed, in (e)(f) is 2 m/s, in (g)/(h) is 0.1 per day.

eddy strength is underestimated in CTRL, particularly over land, although the spatial pattern is well simulated. Similarly, the spatial distribution, including the peaks over the western Pacific and Atlantic are captured (Figs. 3g and 3h).

As a first test of our QSW hypotheses, we calculate Spearman's rank correlation coefficients across the Northern Hemisphere mid-latitudes (15-75N) between climatological values of each of these metrics and the climatological QSW strength. These values, for ERA5 and the CTRL simulation, are shown in the upper section of Table 2. The lower section, which we will discuss more later, presents correlation coefficients for climatological differences in metrics and QSWs strength across different experiments, highlighting the impacts of topography, zonal SST asymmetry, and land surface on QSWs strength changes.





**Table 2.** The spatial correlations between QSW strength and different metrics in ERA5 and experiments calculated between 15°–75°N for DJF only. The values in brackets indicate the variation in correlation coefficients when selecting either the first or last 45 years. The simulations are separated by rows, with ERA5 and CTRL at the top, then the impact of realistic topography (CTRL-NM and ZSST-NM_ZSST), then the impact of realistic zonal SST gradients (CTRL-ZSST and NM-NM_ZSST), and in the bottom section, highly idealized simulations including the impact of land (NM_ZSST-AQUA).

| ERA5/experiments | Stationary wave strength | Stationary wavenumber $Ks$ | Transient eddies strength within $4 \leq Ks \leq 8$ | Eady growth rate |
|---|---|---|---|---|
| ERA5 | **0.24**(±0.08) | **0.42**(±0.01) | **0.85**(±0.00) | **0.39**(±0.00) |
| CTRL | **0.39**(±0.04) | **0.74**(±0.00) | **0.72**(±0.01) | **0.28**(±0.00) |
| CTRL-NM | **0.02**(±0.07) | **0.53**(±0.06) | **0.19**(±0.09) | **-0.23**(±0.01) |
| ZSST-NM_ZSST | **-0.04**(±0.07) | **0.40**(±0.06) | **0.50**(±0.06) | **--0.15**(±0.03) |
| CTRL-ZSST | **0.15**(±0.08) | **0.41**(±0.03) | **0.65**(±0.02) | **0.08**(±0.03) |
| NM-NM_ZSST | **0.27**(±0.02) | **0.39**(±0.03) | **0.63**(±0.05) | **-0.06**(±0.02) |
| IWM-AQUA | **0.19**(±0.05) | **0.43**(±0.13) | **-0.14**(±0.13) | **0.11**(±0.00) |
| HEAT-AQUA | **0.52**(±0.06) | **0.56**(±0.03) | **-0.36**(±0.02) | **0.41**(±0.01) |
| NM_ZSST-AQUA | **-0.11**(±0.01) | **0.37**(±0.03) | **0.47**(±0.12) | **0.48**(±0.07) |

In Table 2, the spatial distribution of QSWs in both ERA5 and CTRL shows relatively strong positive correlations with all metrics. While spatial correlation is a limited method for measuring the association between QSW distribution and other metrics, this result suggests that multiple mechanisms may affect the QSW spatial distribution, and as more stationary forcings are introduced, altering these metrics, their interactions may make it increasingly challenging to isolate and understand the impacts of these stationary forcing on QSWs. By isolating each stationary forcing in turn, and analysing the impacts on the QSWs, we aim to better understand what stationary forcings affect QSWs, and the underlying mechanisms. Spatial distributions of climatological QSWs strength, zonal and meridional wind, stationary wavenumber $Ks$ 4-8 range and transient eddies strength in ERA5 and each experiment can be found in the Appendix, in Figure A3, A4, A5 and A6 accordingly.

To assess the role of internal variability on the correlation coefficients, we separate the 90 years of data into two 45-year periods, and the correlation coefficients in Table 2 do not show significant variation across these different time periods (numbers in parentheses give the variability across the two 45-year periods). In addition, we use more than one pair of experiments to study the impacts of topography and zonal SST gradients, as shown in the sections in Table 2. For example, we use both CTRL-NM and ZSST-NM_ZSST to understand the role of topography, and find strong similarities in the correlation coefficients between these matching pairs, giving further confidence in the robustness of these results.

### 3.3.1 Hypothesis 1: association with stationary waves

To test the first hypothesis, we focus on stationary waves with the wavenumber 4-15 range to match the wavenumber range of the QSWs. We study differences in the strength of these stationary waves between different experiment pairs, and compare to differences in QSW strength (Figure 4). Notably, in most of the experiment comparisons, there is little association between regions where stationary waves are strengthened (weakened) as indicated by solid (dashed) contours, and regions where QSWs are strengthened (weakened) as indicated by red (blue) shading. This is further illustrated by generally weak (although often



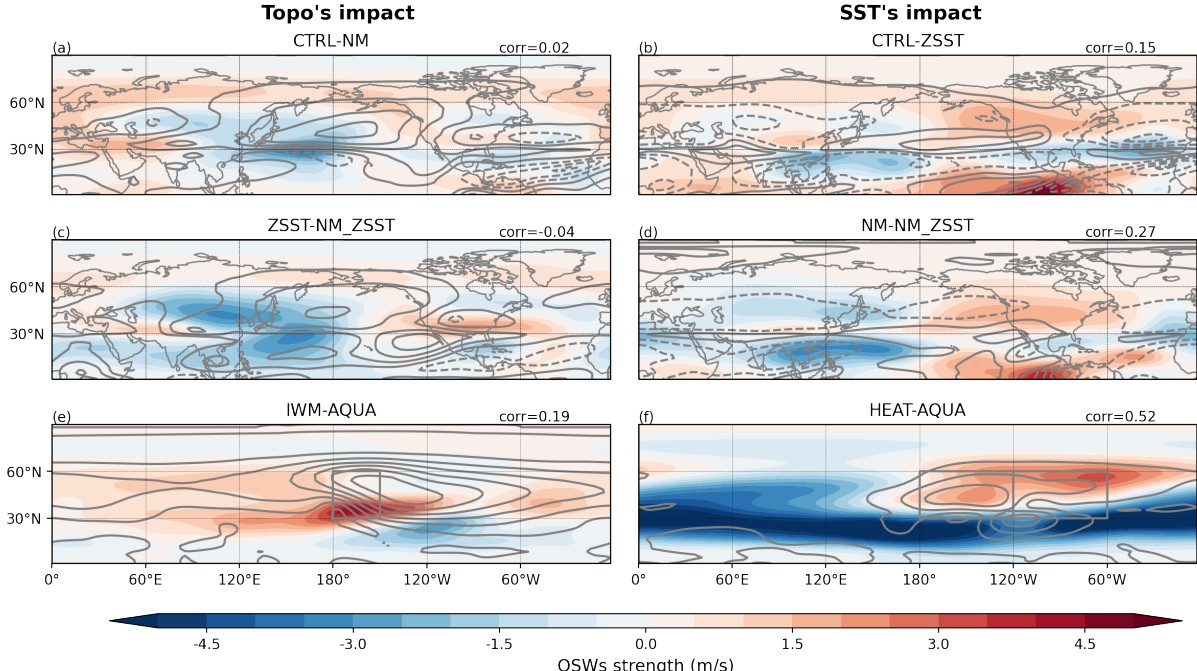

**Figure 4.** Climatological quasi-stationary wave strength (shading) and and wavenumber 4-15 stationary wave strength (contours), with their Spearman's rank correlation coefficients over 15°–75°N attached at top-right corners of subplots. Subplots (a) and (c) display differences between the CTRL and NM, and between ZSST and NM-ZSST. Subplots (b) and (d) show the differences between CTRL and ZSST, and NM and NM-ZSST. Subplots (e) and (f) present differences between IWM and AQUA, and HEAT and AQUA. The left column shows differences illustrating the impact of topography, whilst the right column illustrates the impacts of SST gradients. The contour interval is 1 m/s.

positive) spatial correlation coefficients, given in the top right corner of each panel (and summarized in Table 2), suggesting a weak connection between changes in the stationary waves and the changes in the QSWs; the exception to this is HEAT-AQUA, with a correlation of 0.52 (Fig. 4f). The positive correlations, particularly in response to SST gradients (i.e. Fig. 4b, d, f), which are typically larger than those for changes caused by topography (i.e. Fig. 4a,c,e) suggest there can be a positive relationship between the strength of stationary waves, and the strength of QSWs. However, the generally weak correlations suggest that the hypothesis that QSW strength is associated with stationary wave strength does not primarily explain changes in the spatial distribution of QSWs under conditions other than highly idealized conditions involving only zonal SST asymmetries, i.e. HEAT-AQUA.

While the observed positive correlations between stationary waves and QSWs (Table 2) are consistent with the possibility of wave resonance as a mechanism for high amplitude QSWs, our analysis suggests that it is not a primary factor, particularly in explaining the influence of topography (left column in Fig. 4). It is also possible that the positive correlations are caused by the same factors influencing stationary waves simultaneously affecting QSWs, and not a direct causal relationship between stationary wave strength and QSW strength. For example, QSWs may be linked to zonal wind in different ways in our hypotheses,




and stationary wave strength can also be influenced by zonal wind. Specifically, mechanical forcing from topography interacts with zonal wind, and the strength of stationary waves can vary with the incoming speed and angle of zonal wind as response.

### 3.3.2 Hypothesis 2: stationary wavenumber $Ks$ makes Rossby waves quasi-stationary

We now explore hypothesis 2, starting with the association between QSWs and the stationary wavenumber $Ks$. The highest values of climatological QSW amplitude are typically found where the climatological stationary wavenumber is between 4-8, across a range of very different experiments (Fig. 5). In Figure 5d, representing the AQUA experiment, the relationship between $Ks$ and QSW strength produces a clear line, with little variability about this line and QSW strength peaking within $5 \leq Ks \leq 7$. As more stationary forcings are introduced—such as a single Gaussian water mountain in IWM (Fig. 5c), realistic water mountains in RWM (Fig. 5b), and ultimately in the CTRL (Fig. 5a) experiment—the relationship observed in AQUA becomes increasingly diffuse. However, the "best $Ks$ range" for large QSWs amplitude remains between $4 \leq Ks \leq 8$.

An interesting question arises from Fig. 5: what causes the distinct shape of the heatmaps in Fig. 5, particularly for the AQUA simulation (Fig. 5d)? Since this distinct shape remains relatively consistent regardless of stationary forcings, although more diffuse as more forcings are introduced, it may not be directly related to our main research question of how stationary forcings affect QSWs features; however, understanding this distribution may help understand the underlying mechanisms of QSWs. One potential explanation is the latitude. Both QSW strength and $Ks$ have a similarly clear relationship with latitude, which is more distinct in AQUA and becomes more diffuse, but still recognisable, as more stationary forcings are added (see Figures A7 and A8). This suggests that the beta effect in the stationary wavenumber $Ks$ metric with respect to latitude plays an important role in the total $Ks$, particularly in idealized simulations such as AQUA. With stronger stationary forcings, the background circulation undergoes greater modification, weakening the relationship between latitude and $Ks$ (via the beta effect) and QSW strength. The observed "diffusion" of this relationship when additional stationary forcings are introduced highlights that factors beyond latitude—such as stationary forcings—also influence QSWs distribution, emphasizing the relevance of our research question. In these idealized experiments, as stationary forcings are removed, from A7a to A7d, the latitude range where the stationary wavenumber $4 \leq Ks \leq 8$ shifts equatorward from 20°–50°N to 15°–35°N; a corresponding equatorward shift in the QSW strength peak moves 30°–40°N to 20°–30°N (Fig A8a to A8d), suggests a connection between $Ks$ and QSW strength. To further confirm the relationship between QSW strength and $Ks$, we show the stationary wavenumber $Ks$ distribution overlaid on the climatological QSW distribution in the maps in Figure A5. In Figure A5j (AQUA), the darkest shading aligns precisely with the wavenumber range of 4-8.

We further assess the effectiveness of this stationary wavenumber hypothesis, specifically that QSWs are most likely in regions where $4 \leq Ks \leq 8$, by studying the changes under different stationary forcings. While we do not expect a precise match between Ks of particular values and QSW strength, as QSWs can have a range of both zonal and meridional wavenumbers, we use a proxy for how close the stationary wavenumber is to values between 4 and 8, the absolute difference between the local $Ks$ and a value of 6. Thus, locations with a climatological value of $Ks$ of 5, or 7, would have a value of 1; locations with a value of 3 or 9 would have a value of 3. We then analyze the differences in $|Ks - 6|$ and QSW strength across experiments (Figure 6). If our hypothesis—that smaller values of $|Ks - 6|$ typically correspond to stronger QSWs—is correct, a decrease of $|Ks - 6|$





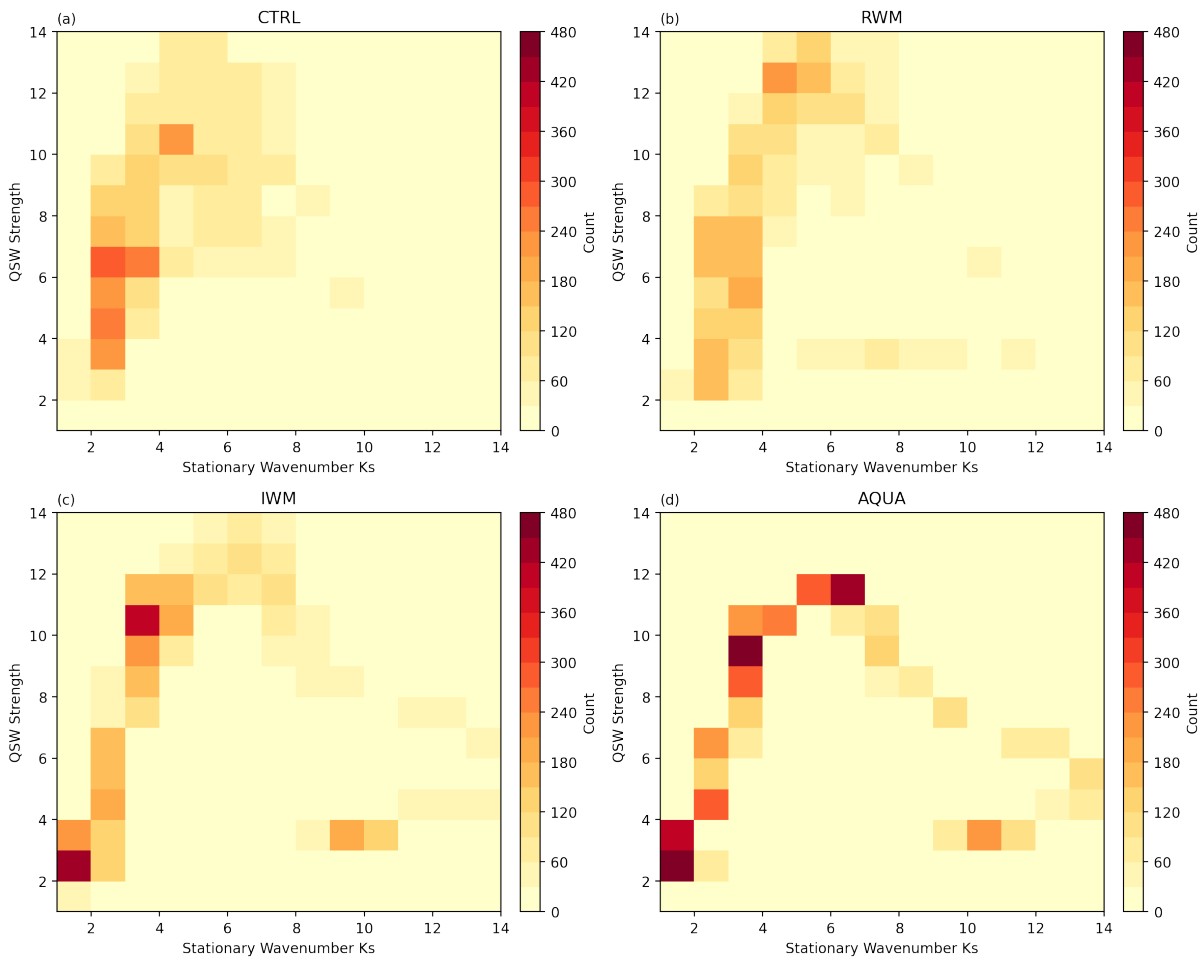

**Figure 5.** Heatmap showing the number of days during DJF that stationary wavenumber $Ks$ is x with a QSW strength of y. (a) shows the statistics in CTRL, (b) for RWM, (c) for IWM and (d) for AQUA

(e.g., in CTRL compared to NM) should coincide with an increase of QSW strength, as the stationary wavenumber moves

closer to 6. To simplify interpretation, we multiply the change of $|Ks - 6|$ by -1, such that, if our hypothesis is correct, then regions where $Ks$ approaches 6 (now shown as positive contours) will align with areas of higher QSW amplitude (positive shading), and the correlation coefficients between $Ks$ and QSW strength will be positive. QSW strength will not peak at a stationary wavenumber of 6 precisely across all regions and experiments, rendering this analysis somewhat quantitative. To ensure the robustness of the stationary wavenumber hypothesis in explaining QSW distributions, we also examined patterns

for choosing stationary wavenumber values of 5 and 7, i.e. calculating differences in $|Ks - 5|$ or $|Ks - 7|$, finding minimal variations in the conclusions.




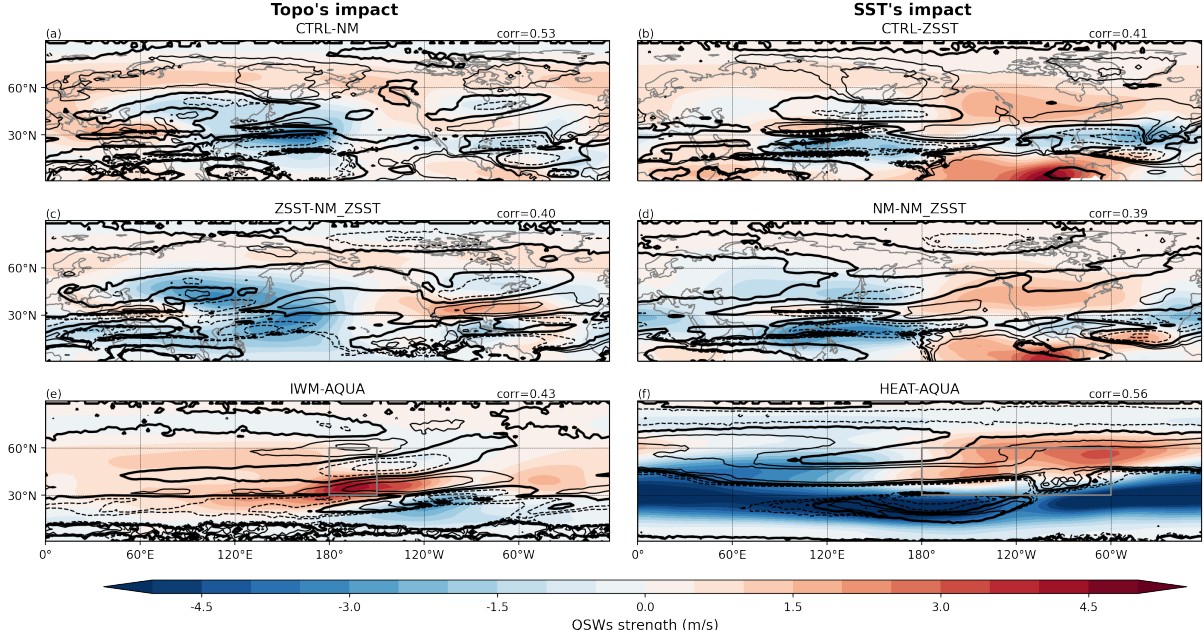

**Figure 6.** How much the stationary wavenumber $Ks$ deviate from $Ks=6$ (|Ks-6|) multiplied by -1(black contours) and QSW strength (shading). The contour interval is 0.5 m/s. Other information is the same as Figure 4.

In Figure 6, in all the experiment pairs regions where $Ks$ becomes closer to 6 tend to have an increased QSW strength, resulting in relatively high positive correlations. One consistent exception to this is downstream of Asian topography in CTRL-NM (Fig. 6a) and ZSST-NM_ZSST (Fig. 6c), where there are regions of $Ks$ closer to 6 (positive black contours), but decreased

climatological QSW amplitude (blue shading). However, over the hemisphere, there are strong positive correlation coefficients in all experiment pairs studying the impacts of topography (0.40 to 0.53). The correlation coefficients in experiment pairs reflecting changes in zonal SST patterns are within a similar range (0.39 to 0.56, right column), suggesting that in most regions, both topography and zonal SST asymmetry may modulate QSWs through their influence on local stationary wavenumber. In Figure 6b, 6d and 6f in the regions where QSWs strength increases the $Ks$ does not change a lot, but still is positive (changed

towards 6). The $Ks$ hypothesis, that high amplitude QSWs are more likely when $Ks$ is closer to 6, is also supported by results from stationary forcings like land-ocean heat contrast and/or land surfaces friction, shown in Figure A2c.

     The analysis above confirms that changes in stationary wavenumber $Ks$ are consistent with the changes seen in the QSWs. We now explore the role of transient eddy strength, as it is also crucial for this hypothesis: under appropriate background flow conditions, transient eddies can decelerate and become quasi-stationary; if there are no transient eddies, the favorable

background condition cannot produce QSWs. Alternatively, if there is a higher Eady growth rate, suggesting transient eddies can grow more easily at a particular location, we may expect more QSWs to form at that location without large transient eddies strength upstream.



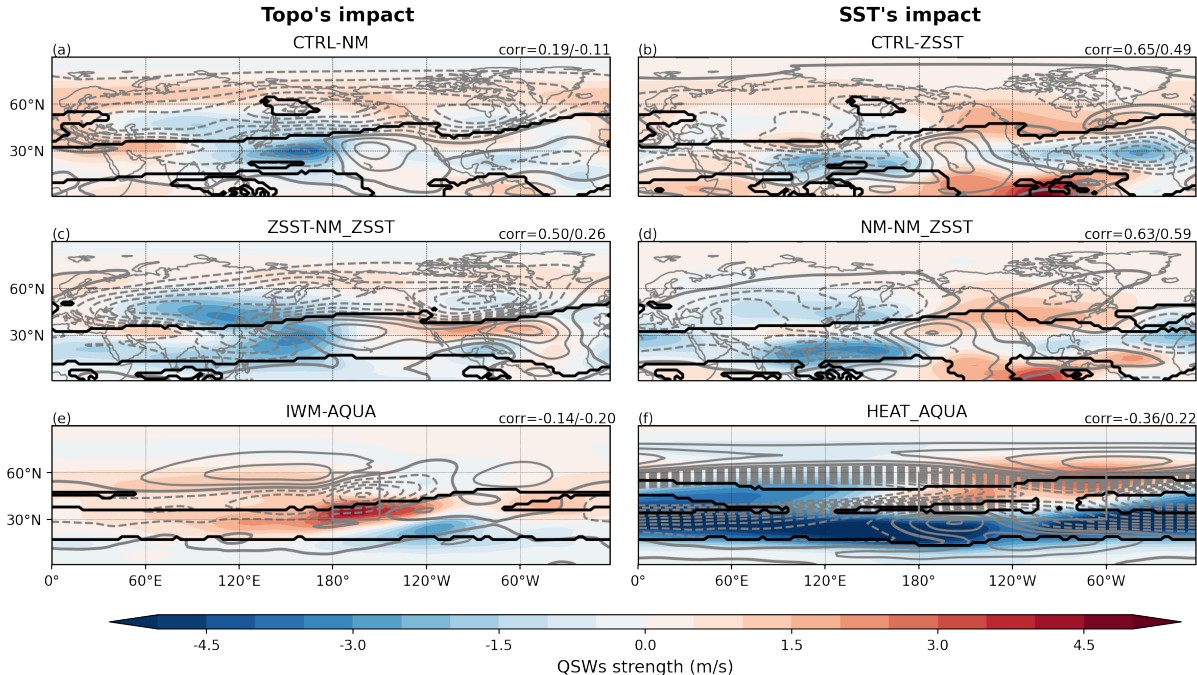

**Figure 7.** The same as Figure 4, except gray contour for transient eddies strength and black contour for stationary wavenumber $4 \leq Ks \leq 8$. The left correlation coefficients are calculated between QSW strength changes and transient eddies strength changes within $4 \leq Ks \leq 8$, while the right correlation coefficients are calculated over all gridboxes between 15°–75°N.

Based on our hypothesis, changes in transient eddies will only have an influence on QSWs if they are within, or close to, a region of $4 \leq Ks \leq 8$. Thus we focus on studying the changes in transient eddies within regions of $4 \leq Ks \leq 8$. In experiments where realistic land is included, changes in the strength of transient eddies are highly correlated with changes in QSW strength within the $4 \leq Ks \leq 8$ range (Figs. 7a-d, values on the left), suggesting an association between transient eddies and QSW strength. However, outside of the region where $4 \leq Ks \leq 8$ (values on the right), the correlations between changes in transient eddy strength and changes in QSW strength are weaker, or negative (Fig. 7a), further evidence for the importance of this $4 \leq Ks \leq 8$ range for QSWs. For idealized topography on an aquaplanet i.e. the water mountain in Figure 7e, changes in the transient eddies strength can not help explain changes in the QSWs distribution at all, suggesting the importance of transient eddies can vary in different configurations, particularly between highly idealized and more realistic configurations.

For experiment pairs studying the impact of SST gradients (right column in Fig. 7), the correlation between transient eddy changes and QSW strength changes is higher in the experiments showing the impact of realistic zonal SST asymmetry i.e. Figs. 7b and 7d, in which QSWs and transient eddies are weakened and strengthened in the same regions. HEAT-AQUA is the only experiment pair where the correlation drops from positive to substantially negative after selecting the $Ks = 4\text{-}8$ range in the Northern Hemisphere, further indicating that stationary wavenumber $Ks$ does not explain QSW strength in the HEAT experiment.




Overall, this analysis indicates that, under realistic configurations, an increase in transient eddy strength is positively correlated with an increase in QSW strength, particularly in regions where $4 \leq Ks \leq 8$. It is worth noting, however, that the
hypothesis suggests a trade-off between QSWs and transient eddies, i.e. QSWs are created by transient eddies becoming stationary - this could lead to a reduction in the amplitude of transient eddies in regions that become more favourable for QSWs. Conversely, in regions where the background conditions are already favourable, an increase in transient eddies could lead to an increase in QSWs. The relationship between changes in transient eddy strength and changes in QSW strength is therefore complex, and the positive correlations found in most experiment pairs suggests that stationary forcing influence on transient
eddies dominates over the conversion of transient eddies into QSWs. In HEAT-AQUA, however, the appearance of a region of $4 \leq Ks \leq 8$ in the high latitudes (see Fig. 6f contours) may be responsible for both the increase in QSWs (see Fig. 6f shading) and the decrease in transient eddies (see Fig. 7f contours), as transient eddies become more quasi-stationary. This may explain the negative correlations between transient eddy changes and QSW changes found for this experiment pair. We analyze the transient eddy strength in all experiments, rather than differences across experiment pairs (shown in Fig. A6). Transient eddy
strength remains high in most areas where QSW strength is strong (compare Figs. A3 and A6), except immediately around Asian topography, as seen in the shading of Figure A3, further strengthening this result that there is a close association between transient eddy strength and QSW strength.

Since transient eddies can locally transition into QSWs under favorable background conditions, any mechanism that modifies wave growth or stability may also directly influence QSWs. We explore this through analysis of the Eady growth rate,
which only shows a strong relationship in experiment pair HEAT-AQUA (Figure 8f). For topography (Fig. 8, left column), the Eady growth rate appears negatively correlated with QSW strength changes over East Asia and the western Pacific, and positively correlated over North America for ZSST-NM_ZSST(Figure 8c), and also locally and downstream of the idealized water mountain (Figure 8e). Accordingly, the hemispheric correlation coefficients vary across experiments in both magnitude and direction, indicating that while Eady growth rate may have a regional impact, particularly for the impact of SST gradients,
it is likely not the primary controlling factor of the changes in QSWs in most configurations.

## 4  Summary and Discussion

In this study, we have examined two hypotheses, involving four variables that may shape the spatial distribution of QSWs. The stationary wavenumber, $Ks$, is the only variable for which there are high correlations with QSW strength in ERA5 and CTRL, and also high correlations in the differences between all experiment pairs we studied (Table 2), suggesting that it can
consistently help explain both the present day QSW distribution, and differences in the QSW spatial distribution between experiments. The transient eddy strength has high correlations in ERA5 and CTRL, and also shows high correlations for zonal SST asymmetry, topography, and land vs aqua-planet experiment pairs, but negative correlations for the experiment pairs illustrating the impacts of idealized topography or idealized SST gradients. This suggests that the transient eddies strength can be important for QSWs, but is not the only factor. The Eady growth rate and stationary strength also show positive correlations
with QSW strength in both CTRL and ERA5, but changes in these variables cannot explain the differences in many experiment



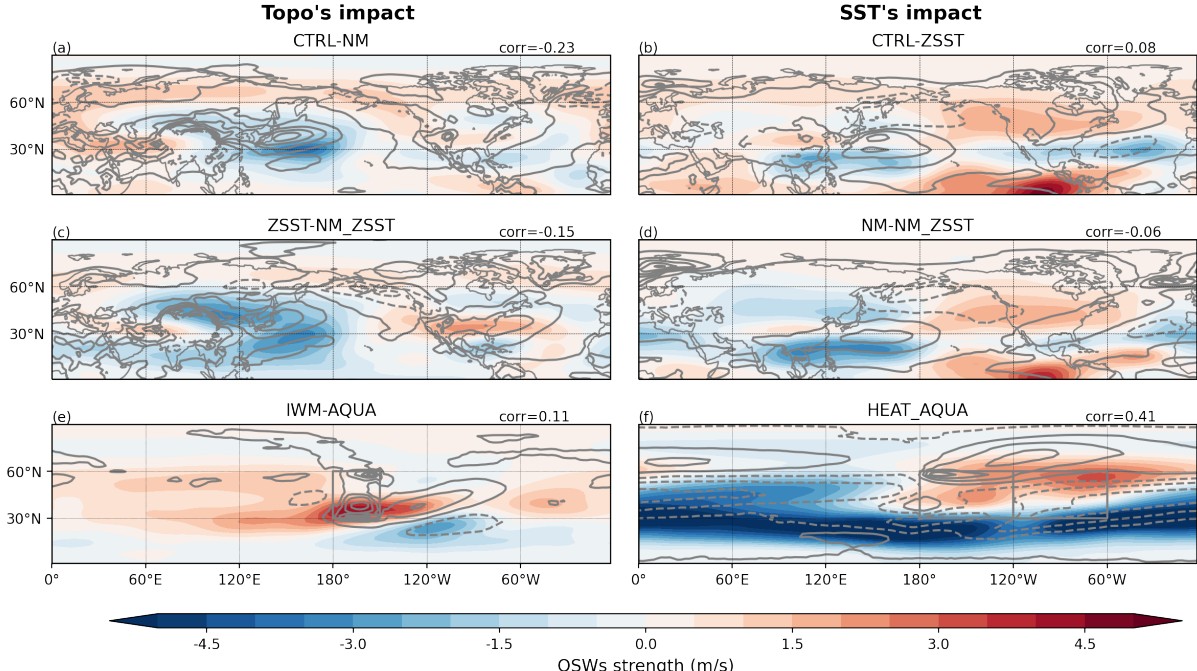

**Figure 8.** The same as Figure 4, except for Eady growth rate at 700 hPa (contour; the interval is 0.1 per day).

pairs. Overall, our results support the hypothesis that background conditions based on stationary wavenumber combined with the presence of transient eddies can explain the spatial distribution of QSWs in many different climate configurations.

It also worth noting that our results show that the correlations in ERA5/CTRL are not always indicative of the correlations in the changes for some factors. Most notably, both the stationary wave strength and the Eady growth rate show a positive

correlation with QSW strength in both ERA5 and CTRL, but show negative, or very low correlations when studying differences between many experiment pairs. This underscores the caution needed when interpreting QSWs distribution in the real world. The complex interplay of forcings in reality makes isolating underlying mechanisms difficult, emphasizing the value of idealized experiments and a model hierarchy in advancing our understanding of QSW dynamics. Whilst 90 years are long simulations, we have tested the robustness of the correlations to the particular selection of year by analyzing half of the full

period - in general, the correlation coefficients remain very similar (see uncertainties in Table 2), confirming the reliability of the results.

In this study, for simplicity, we chose not to employ a latitudinally varying wavenumber range, as in Wolf et al. (2018), and chose to study QSWs at 200hPa. To understand how these choices may have impacted our results and conclusions we compute QSWs for the CTRL, ZSST, and NM simulations at 300hPa, and for different wavenumber ranges (see Fig. 9).

The choice of pressure level between 200 and 300 hPa makes little difference for the CTRL simulation, or the differences between the CTRL and ZSST or NM (compare the top two rows in Fig. 9). Conversely, the choice of wavenumber does have




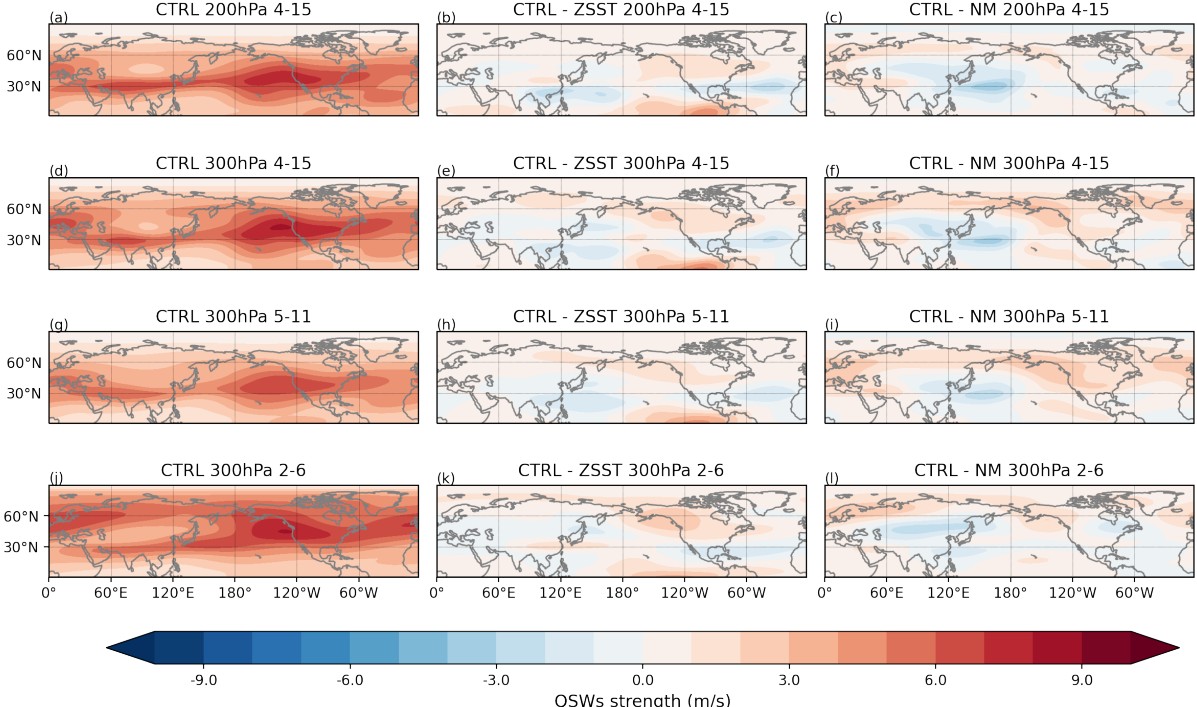

**Figure 9.** This figure illustrates the strength of climatological quasi-stationary waves (QSWs) across various pressure levels and wavenumber ranges. The left column shows QSW strength in CTRL, the middle column shows the difference between CTRL and ZSST, and the right column shows difference between CTRL and NM. The first row shows QSW strength at 200 hPa with wavenumber 4-15, the second row is at 300hPa with wavenumber 4-15, the third row is at 300hPa with wavenumber 5-11, the second row is at 300hPa with wavenumber 2-6.

some impact on the distribution of QSWs in the CTRL simulation (compare Figs. 9g and 9i), which can explain the difference between climatological QSWs in our results (e.g. Fig. 9a) compared to Figure 2b in Wolf et al. (2018). However, the differences between the experiments, i.e., CTRL-ZSST and CTRL-NM, show little sensitivity to the choice of wavenumber (compare Fig.

9e with 9h and 9k, and similarly 9f with 9i and 9l), suggesting that these choices likely have relatively little impact on our main results and conclusions.

To assess the robustness of our results to the definition of 'quasi-stationary' that we use, we also calculated QSW strength using both a 15–30-day Lanczos band-pass filter and a 30-day Lanczos low-pass filter. The results (summarized in Table 3) indicate that the relationship between QSW changes and variations in the selected metrics remains relatively consistent

across these distinct filtering approaches, particularly the strong correlations with $Ks$ and transient eddies, for experiment pairs with realistic land. However, experiment pairs using the 15–30-day band-pass filter exhibit greater differences from the 15-day low-pass filter (given in Table 2) than from the 30-day low-pass filter, especially in correlations with $Ks$ and transient eddy strength. This suggests potential differences in QSW behavior across different timescales. The two experiment pairs with the lowest correlation coefficients with Ks for 15-30 day bandpass QSWs are NM-NM_ZSST, and IWM-AQUA. If we




**Table 3.** The spatial correlations between QSW strength and different metrics in ERA5 and experiments within 15°–75°N are presented. The left values represent the correlations using QSWs calculated with 15–30-day band-pass filtered meridional winds. The right values represent the correlations using QSWs with 30-day low-pass filtered meridional winds.

| ERA5/experiments | Stationary wave strength | Stationary wavenumber $Ks$ | Transient eddies strength within $4 \leq Ks \leq 8$ | Eady growth rate |
|---|---|---|---|---|
| ERA5 | 0.17/0.27 | 0.41/0.43 | 0.88/0.83 | 0.41/0.38 |
| CTRL | 0.34/0.41 | 0.72/0.74 | 0.78/0.69 | 0.25/0.29 |
| CTRL-NM | 0.20/0.15 | 0.47/0.45 | 0.44/0.13 | -0.17/-0.26 |
| ZSST-NM_ZSST | 0.04/0.12 | 0.25/0.38 | 0.30/0.50 | -0.29/-0.23 |
| CTRL-ZSST | 0.19/0.11 | 0.40/0.37 | 0.87/0.47 | 0.30/-0.07 |
| NM-NM_ZSST | 0.08/0.14 | 0.17/0.38 | 0.56/0.55 | -0.32/-0.15 |
| IWM-AQUA | -0.41/0.51 | 0.09/0.47 | -0.19/-0.05 | -0.08/0.22 |
| HEAT-AQUA | 0.45/0.52 | 0.35/0.61 | 0.03/-0.42 | 0.56/0.38 |
| NM_ZSST-AQUA | 0.00/-0.12 | 0.26/0.35 | 0.47/0.40 | -0.08/0.42 |

increase the timescale range to 15-45 days, these correlation coefficients increase from 0.17 to 0.26 and from 0.09 to 0.18 respectively, illustrating that the higher correlations in the 30 day band-pass filter are not solely from very low frequency signals. Nonetheless, the consistently high correlations with $Ks$ and transient eddy strength in CTRL–ZSST and CTRL–NM, along with the positive correlations in ZSST–NM_ZSST and NM–NM_ZSST, suggest that quasi-stationary waves on both shorter (15–30 days) and longer (30–90 days) timescales share similar characteristics and governing mechanisms under semi-

realistic conditions. In more idealized experiment pairs, different mechanisms may contribute to QSWs in varying proportions depending on their timescale. These findings broaden our understanding of QSW dynamics.

Overall, our results provide the most support for our second hypothesis - that transient eddies can become quasi-stationary under favorable background conditions, which can be identified through the stationary wavenumber, $Ks$. This does not, however, prove that the mechanisms proposed in hypothesis 1 are completely invalid, or that non-linear processes do not also

play a role in QSWs. Considering hypothesis 1, related to the mechanism of quasi-resonant amplification (QRA) proposed by Petoukhov et al. (2013), this mechanism further requires the existence of a waveguide and the same phase for stationary waves and transient eddies. Our research tests only one aspect, the connection to stationary waves, and our conclusions are drawn solely from climatological statistics. For analyzing extreme events, the QRA framework may still provide a useful approach.

It could also be argued that $Ks$ could impact both quasi-stationary and stationary waves similarly, as we find positive

correlations between QSWs strength and $Ks$, and $Ks$ is derived for stationary waves originally; however, the experiment pairs that show the strongest positive correlation between QSWs and $Ks$ (CTRL - NM; CTRL-ZSST) notably do not show a strong correlation between QSWs and wavenumber 4-15 stationary wave strength in Table 2. Indeed the only experiment pair with a strong positive correlation between QSWs and wavenumber 4-15 stationary waves (HEAT-AQUA) has a nearly zero correlation between QSWs and $Ks$. Thus, stationary waves with wavenumbers 4-15 are not primarily affected by stationary wavenumber

in the same way as QSWs.

In Hypothesis 2, we have emphasized the importance of background conditions affecting QSW distribution using a linear framework in different ways. However, transient eddies can become quasi-stationary via other mechanisms and/or including



non-linear interactions help sustain quasi-stationary waves through other mechanisms (e.g. Nakamura and Huang, 2018; Ma and Franzke, 2021; Karoly et al., 1989). As an example, Nakamura and Huang (2018) show how zonally averaged zonal wind, group velocity (determined by wavenumber), and stationary waves collectively constrain the local flux capacity of the atmosphere to transport wave energy. When the flux capacity is exceeded, circulation anomalies can become quasi-stationary. Given our results showing that changes in zonal mean zonal winds and stationary waves do not have a simple direct relationship to changes in QSWs, this suggests that no single factor in the three variables important for the flux capacity acts as a dominant control on QSW distribution. Future work is needed to explore whether flux capacity is useful to explain changes in QSWs, and how the three factors collectively contribute.

Our results show that the wavenumber $Ks$ and transient eddies can explain a significant proportion of the spatial distribution of QSWs under many substantially different climates; however, this does not address the question of QSWs' energy sources. Further analysis is needed to quantify and understand the energy sources and transfers involved in QSW dynamics. It would also be interesting to further explore the impact of zonal SST asymmetries in some cases, e.g. under different phases of internal variability or in a warming climate.

## 5 Conclusions

In this research, we conducted eight CAM6 simulations with prescribed SSTs to understand the impact of stationary forcings, topography, zonal SST gradients, and land surface, on the characteristics of quasi-stationary waves. Comparison of our CTRL experiment with ERA5 reanalysis confirms that QSWs, and the associated variables that we study, are reasonably well replicated by the CAM6 model. Our findings reveal that the presence of topography, land, and/or SST gradients tends to extend the duration of QSW events and increase QSWs zonal mean amplitude in the extratropics, as well as having a strong influence on the spatial distribution of climatological QSW amplitude.

Changes in QSWs in response to stationary forcings are primarily associated with changes in transient eddies strength and the background flow conditions, as described by the local stationary wavenumber $Ks$, supporting a barotropic linear understanding of QSWs. Association with stationary waves, despite relatively strong correlations in ERA5 and CTRL, only help explain changes in QSW strength in very idealized experiments, with idealized SST heating on an aquaplanet. When QSWs are defined on different timescales, we find similar mechanisms are important, i.e. $Ks$ and transient eddies, but perhaps with different proportions. This work helps further our understanding of QSWs, and their underlying mechanisms.

*Data availability.* ERA5 data were downloaded from the Copernicus Data Store (CDS). Idealized experiments data can be asked from the authors.

## Appendix A: Additional figures





**Figure A1.** The same as Figure 1, except for statistics in Europe (0-60°E).





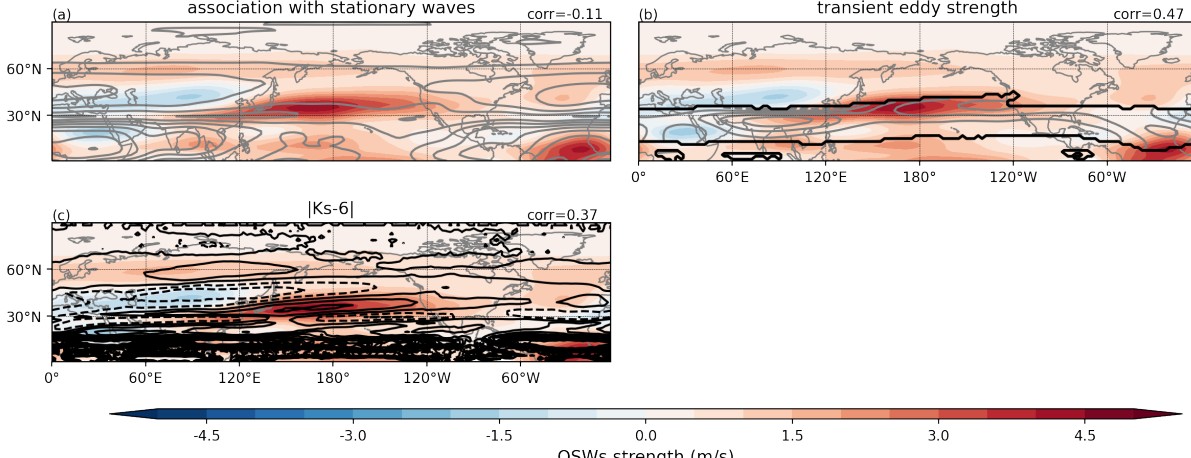

**Figure A2.** The difference between NM-ZSST and AQUA showing the impact of land surface. Climatological QSW strength (shading) and climatological zonal wind gradient (contour) in (a), climatological transient eddies strength (contour) in (b), How much the stationary wavenumber $Ks$ deviated from $Ks$=6 (contour) in (c).



**Figure A3.** Climatological QSW amplitude in winter (shading) and climatological zonal wind (contour) in ERA5 (a) and idealized experiments - CTRL (b), ZSST (c), NM (d), NM-ZSST (e), RWM (f), IWM (g), HEAT (h), and AQUA (i). The line spacing is 4 m/s, solid lines represent positive anomalies whilst dashed lines represent negative anomalies and zero contour. The ERA5 reanalysis time range is from 1940 to 2014, and the time range of idealized experiment is 100 years.





**Figure A4.** Climatological zonal wind in winter (shading) and climatological meridional wind (contour). Other information is the same as Figure A3



**Figure A5.** Climatological QSW strength in winter (shading) and stationary wavenumber $Ks$=4 and $Ks$=8 (contour). Other information is the same as Figure A3





**Figure A6.** This figure shows climatological transient eddy strength. Other information is the same as Figure A3





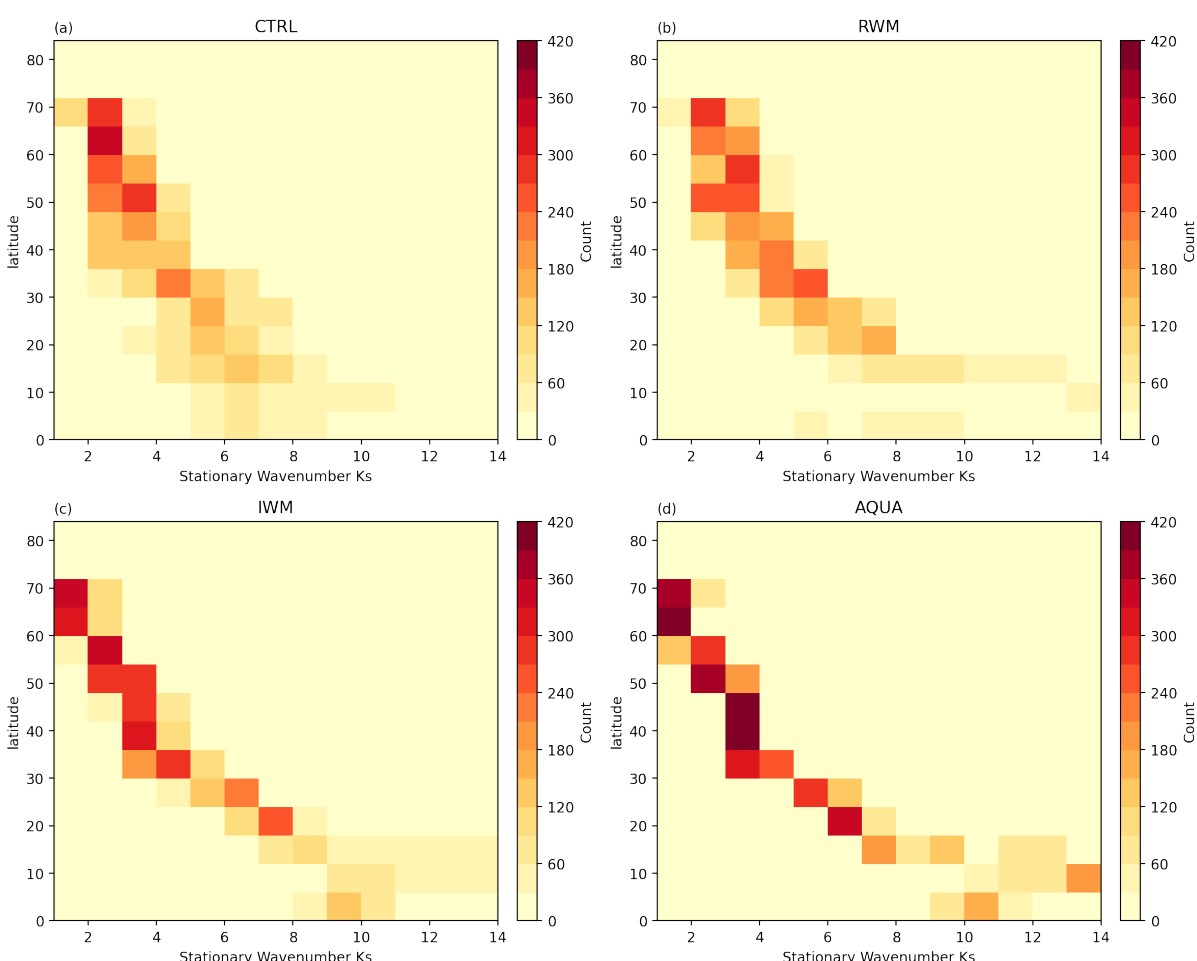

**Figure A7.** Heatmap shows the count of cases that stationary wavenumber $Ks$ is $x$ at latitude $y$. (a) shows the statistics in CTRL, (b) for RWM, (c) for IWM and (d) for AQUA

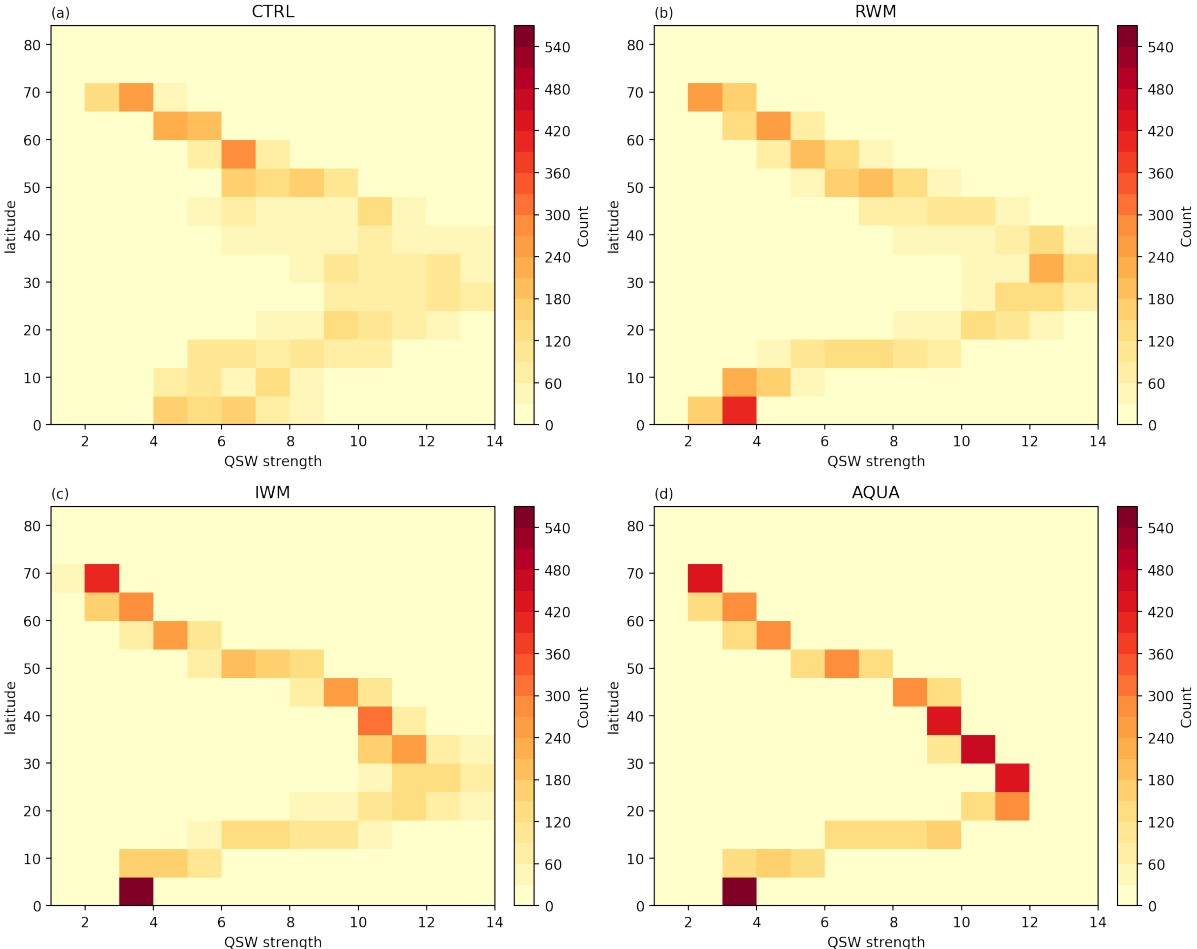

**Figure A8.** Heatmap shows the count of cases that QSW strength is $x$ at latitude $y$. (a) shows the statistics in CTRL, (b) for RWM, (c) for IWM and (d) for AQUA

*Author contributions.* CF designed the experiments, did the analysis, created the plots, and wrote the paper draft. RHW supervised the study and revised the paper.

*Competing interests.* The authors declare no competing interests

*Acknowledgements.* This research was supported in part through computational resources and services provided by Advanced Research Computing at the University of British Columbia and Compute Canada. CF would like to thank the tremendous help from Dr. Brian Dobbins and ARC Sockeye people (mostly Dr. Ken Bigelow) during installing CESM2. CF would also like to thank the discussion with Prof. Tiffany




Shaw about the definition of QSWs during Rossbypalooza. CF was supported by NSERC Discovery Grant [RGPIN-2020-05783]. CF would like to acknowledge ChatGPT in correcting the grammar for the manuscript.



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
