# Peer review of "The Role of Topography, Land and Sea Surface Temperature on Quasi-Stationary Waves in Northern Hemisphere Winter: Insights from CAM6 Simulations"

_EGUsphere, 2025_

## Referee Comment (RC1)

**Review of WCD-2024-3435**
**"The Role of Topography, Land and Sea Surface Temperature on Quasi-Stationary Waves in Northern Hemisphere Winter: Insights from CAM6 Simulations"**

by
Cuiyi Fei & Rachel H. White

**Recommendation: Minor Revisions**

Fei & White (2025) examine the effect of stationary forcings (e.g., topography, SSTs) on quasi-stationary wave (QSW) characteristics using a set of 100-year CAM6 simulations with varying SSTs, topography, and land-sea distribution. The authors find that background conditions associated with stationary wave number and transient eddies explain most of the variance in spatial QSW distributions.

Overall, the manuscript is well organized and well written. The authors provide a thorough introduction with a nice overview of how quasi-stationary waves relate to other atmospheric and environmental features/processes. The methods, overall, appear sound and in line with previous work, and I appreciate the organized approach to isolating different stationary forcings with the model simulations. The results are also generally well presented—the authors do well to walk the reader through fairly detailed comparisons. I also appreciate the inclusion of various sensitivity tests throughout to test robustness of presented results.

There are a few places in need of additional information or clarification after which I believe this manuscript to be suitable for publication in *Weather and Climate Dynamics*.

Comments

L50–52: In addition to ENSO, how would convection related to the MJO impact QSWs?

L60: Remove "this" between "in" and "highly".

L100: Replace "resolution" with "grid spacing" as the two are not synonymous (i.e., features are resolved at 4–6 times the grid spacing).

L100–101: Given the time spent in the Introduction on diabatic heating influences, it would be worth a short discussion (perhaps in Section 4) on how this coarse grid spacing (and therefore, lack of diabatic processes) may influence your results.

L122–125: How was the spatial extent for the HEAT SST anomalies chosen?

L155, L162: Suggest putting two equations on separate lines.

L176: Should "CESM2" be "CAM6"?

Fig. 1: I wonder if making each line slightly transparent would help the reader with interpretation, especially with identifying areas of overlap between experiments.

Fig. 2: Missing y-axis labels.

Fig. 2: Missing reference to panels (d), (e), and (f) in figure caption.

Fig. 3: Suggest including a pattern correlation coefficient analysis here to strengthen your ERA5-CNTL comparisons more objectively.

L260: Specify which section "later" refers to.

L279–281: Remove parenthetical sentence structure and split into two sentences to increase readability.

L316: Add "Figure" before "A7a to A7d".

L320: Suggest reordering Appendix figures so they are referenced in order in the text.

L326: Should the last "value" be "Ks value"?

Fig. 6: Suggest smoothing contours to help reduce noise and highlight signals.

L355: Is this really true for (a)? The correlation is only 0.19.

L357–358: It appears the right correlations are weaker in (a) and (c) but right and left correlations in (b) and (d) are fairly similar. Please add additional discussion to explain this difference in result.

L440: Acronym already defined in L70.

Section 5: The conclusions would benefit from bringing the presented results back to some of the implications of QSWs discussed in the Introduction. For example, what do these identified relationships between QSWs and certain variables mean for, say, forecasting QSWs and their effects on weather conditions?

---

## Author Comment (AC1)

I have copied the major comments of the reviewer in black, along with my responses in blue.

Fei & White (2025) examine the effect of stationary forcings (e.g., topography, SSTs) on quasi-stationary wave (QSW) characteristics using a set of 100-year CAM6 simulations with varying SSTs, topography, and land-sea distribution. The authors find that background conditions associated with stationary wave number and transient eddies explain most of the variance in spatial QSW distributions.

Overall, the manuscript is well organized and well written. The authors provide a thorough introduction with a nice overview of how quasi-stationary waves relate to other atmospheric and environmental features/processes. The methods, overall, appear sound and in line with previous work, and I appreciate the organized approach to isolating different stationary forcings with the model simulations. The results are also generally well presented—the authors do well to walk the reader through fairly detailed comparisons. I also appreciate the inclusion of various sensitivity tests throughout to test robustness of presented results.

Thank you very much to the reviewers for their thoughtful and helpful feedback. We greatly appreciate the recognition of the contributions and clarity of our work.

There are a few places in need of additional information or clarification after which I believe this manuscript to be suitable for publication in Weather and Climate Dynamics.

Comments L50–52: In addition to ENSO, how would convection related to the MJO impact QSWs?

Thanks for your suggestion. We will include a discussion of the MJO in the revised manuscript. Generally, both ENSO and MJO influence QSWs through convection, with no fundamental difference, since the heating released by convection is considered a source of Rossby waves, regardless of the source of the convection.

L60: Remove "this" between "in" and "highly".

Thanks for your suggestion. We will remove the word 'this' in the revised manuscript.

L100: Replace "resolution" with "grid spacing" as the two are not synonymous (i.e., features are resolved at 4–6 times the grid spacing).

Thank you. We will replace resolution to make the terms consistent.

L100–101: Given the time spent in the Introduction on diabatic heating influences, it would be worth a short discussion (perhaps in Section 4) on how this coarse grid spacing (and therefore, lack of diabatic processes) may influence your results.

Thanks for your suggestion. We will add more discussion about the impact of the coarse grid spacing in the discussion.

L122–125: How was the spatial extent for the HEAT SST anomalies chosen?

To replicate the temperature contrast between Siberia and the western Pacific Ocean, we defined the SST anomaly region to span 60 degrees of longitude in the midlatitudes, mimicking the order of magnitude longitudinal extent of an ocean basin or continent. We will add one sentence in the revised manuscript to explain this.

L155, L162: Suggest putting two equations on separate lines.

Thanks for your suggestion. We will put two equations on separate lines in the revised manuscript.

L176: Should "CESM2" be "CAM6"?

Thanks for your suggestion. We will replace CESM2 by CAM6 in the revised manuscript.

Fig. 1: I wonder if making each line slightly transparent would help the reader with interpretation, especially with identifying areas of overlap between experiments.

Thanks for your suggestion. That's a great suggestion. We will make the lines partially transparent to improve clarity when lines overlap.

Fig. 2: Missing y-axis labels.

Thank you. This figure will be removed based on the other reviewer's suggestion.

Fig. 2: Missing reference to panels (d), (e), and (f) in figure caption.

Thank you. This figure will be removed based on the other reviewer's suggestion.

Fig. 3: Suggest including a pattern correlation coefficient analysis here to strengthen your ERA5- CNTL comparisons more objectively.

Thanks for your suggestion. We will add the pattern correlation coefficients to the upper right corner of the CTRL experiment figures later.

L260: Specify which section "later" refers to.

Thanks for your suggestion. The "later" refers to subsections 3.3.1 and 3.3.2 as a whole; we will clarify this in the revised manuscript.

L279–281: Remove parenthetical sentence structure and split into two sentences to increase readability.

Thanks for your suggestion. We will remove the parenthetical content and use another sentence to illustrate it in the revised manuscript.

L316: Add "Figure" before "A7a to A7d".

Sorry for that and thanks for noting this. We will remember to add 'Figure' wherever necessary in the revised manuscript.

L320: Suggest reordering Appendix figures so they are referenced in order in the text.

Thanks for your suggestion. We will double check and reorder the Appendix figures in the revised manuscript.

L326: Should the last "value" be "Ks value"?

Thanks for your suggestion. We will specify the 'Ks value' explicitly, and we will ensure that all values are clearly defined in the revised manuscript.

Fig. 6: Suggest smoothing contours to help reduce noise and highlight signals.

Thanks for your suggestion. We will smooth contours in the revised manuscript.

L355: Is this really true for (a)? The correlation is only 0.19.

Thanks for pointing this out. We will be more careful about our description in the revised manuscript.

L357–358: It appears the right correlations are weaker in (a) and (c) but right and left correlations in (b) and (d) are fairly similar. Please add additional discussion to explain this difference in result.

Thanks for your suggestion. We will discuss the potential distinct impacts of topography and diabatic heating in the revised manuscript, which also aligns with our planned response to a related comment from another reviewer.

L440: Acronym already defined in L70.

Thanks you. We will remove the second reference to the acronym in the revised manuscript.

Section 5: The conclusions would benefit from bringing the presented results back to some of the implications of QSWs discussed in the Introduction. For example, what do these identified

relationships between QSWs and certain variables mean for, say, forecasting QSWs and their effects on weather conditions?

Thanks for your suggestion. We certainly consider that aspect—for instance, in relation to midwinter suppression. Although we have not yet begun analyzing it due to PhD graduation commitments, we will include this discussion in the revised manuscript.

---

## Author Comment (AC2)

I have copied the major comments of the reviewer in black, along with my responses in blue.

General Comments:

The paper presents an investigation into the Quasi Stationary Wave Strength and its dependence on surface forcing and internal dynamical factors. The investigation is performed using eight 100 year simulations from the CAM6 model. I believe the subject is of interest to the community and the set of simulations has been systematically/logically designed to probe the relationship between different surface forcings and Quasi Stationary Waves.

I found the paper quite difficult to read and spent a lot of time flicking back and forth between various sections to work out what was plotted in the various figures and why, so feel it would benefit from a careful revision to the text to make it more explicit in terms of analysis and interpretation and some restructuring to make the main scientific questions/hypotheses and methods clear. It is possible the paper is attempting to cover too much material so there are many multi-panel figures, which require a lot of explanation. Reducing the scope of the paper would in my view make it much clearer and stronger.

We thank the reviewer for taking the time to review our paper. Their comments and suggestions will significantly improve the manuscript. In response to this comment, we will remove the current Section 3.2 (on the role of zonal-mean zonal wind and meridional wind) to improve the readability and reduce the scope. We will also add more analysis to illustrate the distribution of quasi-stationary waves (QSWs) through temporal analysis, as suggested by the reviewer. Additionally, we will outline and summarize the hypotheses, analysis, and reasoning/conclusions as bullet points to better highlight the core message of the paper.

The main body of the paper is devoted to testing two hypothesis:

1 The spatial distribution of quasi-stationary waves is governed by that of stationary waves.

2 The spatial distribution of quasi-stationary waves is governed by that of dynamic factors such as Eady growth rate and local stationary wavenumber and transient wave strength. These hypotheses appear in the results section, but I think they should really appear in the introduction motivated by/linked to the discussion of literature. The methods section can then more clearly explain how the methods are designed to address the hypotheses

Thanks for your suggestion. We will fit the hypotheses in the introduction as suggested. In addition to these two main hypotheses on the distribution of QSWs, we also explore the impact of stationary forcings on the duration of QSWs, and plan to keep this analysis in the manuscript, as we believe it's an important result for this work.

The partition between stationary, quasi-stationary and transient waves defined by the authors is as follows:

- Stationary waves: A day of year climatological mean.

- Quasi-stationary waves: 15 day low pass filtered anomalies the climatology

- Transient waves: 15 day high pass filtered anomalies to the climatology

All three of these are converted to wave-envelope amplitude (as in Zimin et al 2003) including only wavenumbers 4-15.

A major methodology employed in the paper is comparison of climatological mean spatial patterns of quasi-stationary wave amplitude to that of stationary waves and transient waves as well as to climatological mean spatial patterns of Eady growth rate and stationary wavenumber. So a number of the conclusions hang on the assumption that a similar climatological mean pattern implies a significant relationship between quantities/lack of a similar pattern implies no relationship. I think this method is employed in all figures except fig 1and fig 5 which makes use of time variation in the data. The robustness of this basic assumption requires some discussion in the text - it isn't clear from what is written why this is a useful method and what its possible limitations are. The authors could also consider including more results which more carefully examine the relationship between stationary wavenumber and Quasi-stationary wave strength by utilising information about time covariation of these two quantities - this to me would add weight to the argument that the two a closely linked.

Thank you for your important suggestion. We will add more discussion of this method and its limitations. We will add new temporal analyses, including lead-lag composites between QSW strength, stationary wavenumber Ks, and transient eddy strength, and composites of Ks during QSW and non-QSW days. Based on our current results, we find a clear lead-lag relationship between QSW strength and transient eddy strength, although the pattern varies across regions and experiments.

The main conclusion of the paper is that stationary wavenumber has a clear association with QSW strength suggesting a simple barotropic interpretation. This is an interesting conclusion, but the more clarity is needed in the explanations of results to justify this.

Thank you for your suggestion. We will revise the wording in the next version of the manuscript. Our argument is based on the fact that the stationary wavenumber Ks is derived from the dispersion relation for barotropic Rossby waves. Therefore, even if some nonlinearity is allowed in the process, the QSWs should remain (equivalent) barotropic throughout their lifecycle. We will include a more detailed explanation of our results and

discussion of this point in the Discussion section to link the conclusions to the results more clearly.

Specific Comments:

- There is a long introduction which covers well the literature in the area. This ends with a short paragraph about the scientific question/hypothesis the authors' work seeks to address. The hypothesis in this final paragraph is too vague and doesn't really explain what the authors aims are and how these fit into the literature described prior to this.

Thanks for your suggestion. We will add more introduction about how these stationary forcings affect circulation, and then affect QSWs based on literature. We will also specify the hypotheses at the end of introduction more clearly as suggested in the next comment.

- Two hypotheses are given in section 3.3. The bulk of the paper seems to be aimed at testing these, so perhaps it would make sense to include these in the introduction and motivate them from the literature - why are these important and novel hypotheses to test. Then to include some discussion of the methods used to test these hypotheses in the methods section.

Thanks for your suggestion. We will state the hypotheses and design of analysis at the end of introduction.

- section 3.1 line 170 What does "frequency of all events is fixed" mean - explain this more clearly. Do you mean you are defining an amplitude threshold for an "event" such that the total number of events is the same between the different simulations and reanalysis?

Thanks for your suggestion. Indeed – in each dataset (reanalysis or simulation), we select the strongest 90 QSW events, corresponding to the 90 analyzed years, to ensure a fixed frequency of, on average, one QSW event per year. However, in response to other comments about reducing the panels in this figure, we will remove this analysis from the main paper, and keep it only in the Appendix. This analysis shows that our results on duration are not sensitive to how we select events in the different simulations.

- Main conclusions of Figure 1 seem to be that the model is biased towards shorter duration events, including stronger forcing increases the duration of events. You could consider whether these points could be demonstrated with a simpler figure with fewer panels. It would also be good to comment on whether these conclusions would be expected from what we know about climate models/other modelling studies or are specific to this particular model.

Thank you for your suggestion. We will move the fixed frequency results to the Appendix. We will also expand the discussion on the performance of the CTRL experiment, including its limitations—such as the shortcomings of using prescribed SSTs, and connect these to

- section 3.2 you state that amplitude of meridional wind is used as proxy for stationary wave amplitude. This implies you are calculating the zonal mean of magnitude of the time mean meridional wind, not the time-mean zonal-mean of the magnitude of the meridional wind which would include transient waves. However you don't state clearly in the text what is calculated.

Thanks for your suggestion. We've removed this section now.

- line 257: "As a first test, we calculate Spearman rank correlation coefficients". This appears to be the main test that is used in the paper to link QSWs to the other quantities.

Thank you for your comment. Our distinction was between the first test – correlation on the absolute fields from simulations, and the second test – correlation on the differences between experiments. We will now also include the temporal analysis, as suggested by the reviewer, and so will remove the language around 'first test'.

- Figure 3. Panels a and b are very clear, but it would be good to state the contour interval where they are described in the caption rather than at the end.

Thanks for your suggestion. We will include the contour interval in the revised manuscript.

- The effects of different types of surface on QSWs are investigated by examining differences in climatological means between different pairs of simulations, rather than just looking at anomalies from the control simulation. This makes the results quite difficult to follow, my question is whether this complication is necessary or anomalies from the control could be used instead or whether the anomalies are necessary at all - i.e. could you make the same points just by looking at the fields themselves and comparing? This might simplify the analysis. If it is necessary to look at these anomalies as in the current paper to make the points, then it would be good to give more explanation to help the reader understand what they are learning from the different pairings.

Thank you for your suggestion. As mentioned above, we compare differences in climatological means between different pairs of simulations because (1) each metric shows some degree of correlation with QSW strength when comparing climatological values directly, but the differences do not necessarily (see Table 1), and (2) we aim to understand whether different stationary forcings (e.g. topography vs diabatic heating) influence QSWs in distinct ways. We will add further discussion to clarify the reasoning behind this methodology, and on the potentially different impacts of diabatic heating and topography on QSWs.

- A significant portion of the paper addresses the link between stationary wavenumber and quasi-stationary wave amplitude. This to me was the more interesting part (particularly fig

5), however the method for calculating stationary wave number is missing some key details: the authors state it is calculated from zonal wind, but don't specify whether this is a single level a vertical integral. In figures 3c and 3d, I can not determine what the contours are showing. At first I assumed from the caption/title that these were the stationary wavenumber with only values 4 to 8 shown, however the contours range from 2 to 10 so is this just stationary wavenumber?, if so why have 4-8 in the title - please provide a clear explanation.

Thanks for your suggestion. We calculate the stationary wavenumber Ks at 200 hPa, the same as the level where QSWs strength is calculated. We will add it in the revised manuscript. Thanks for pointing out the incorrect title - we will correct it in a revised manuscript, and add details in the caption about the contours.

- Figure 5 is very interesting and appears to show a clear link between the Ks and the QSW amplitude. However it is not clear from the caption of text what is actually plotted here. The count refers to the number of days in DJF which is clear and one assumes this sums up to 90xlength of simulation in years. But how do the authors decide what the QSW strength and stationary wavenumber are for a given day - is this a spatial mean value?

Thank you for your suggestion. The caption is incorrect—it should refer to the count of grid points, not days. The values of both stationary wavenumber Ks and QSW strength shown are climatological values. However, as per your earlier suggestion, we will add more temporal analyses. In those, the count will be based on grid points per day, thereby incorporating time information.

---

## Author Response (AR1)

I have copied the major comments of the reviewer in black, along with my responses in blue.

**Reviewer 1:**

Fei & White (2025) examine the effect of stationary forcings (e.g., topography, SSTs) on quasi-stationary wave (QSW) characteristics using a set of 100-year CAM6 simulations with varying SSTs, topography, and land-sea distribution. The authors find that background conditions associated with stationary wave number and transient eddies explain most of the variance in spatial QSW distributions.

Overall, the manuscript is well organized and well written. The authors provide a thorough introduction with a nice overview of how quasi-stationary waves relate to other atmospheric and environmental features/processes. The methods, overall, appear sound and in line with previous work, and I appreciate the organized approach to isolating different stationary forcings with the model simulations. The results are also generally well presented—the authors do well to walk the reader through fairly detailed comparisons. I also appreciate the inclusion of various sensitivity tests throughout to test robustness of presented results.

Thank you very much to the reviewers for their thoughtful and helpful feedback. We greatly appreciate the recognition of the contributions and clarity of our work.

There are a few places in need of additional information or clarification after which I believe this manuscript to be suitable for publication in Weather and Climate Dynamics.

Comments L50–52: In addition to ENSO, how would convection related to the MJO impact QSWs?

Thanks for your suggestion. We have included a discussion of the MJO in the revised manuscript. Generally, both ENSO and MJO can influence QSWs through convection, with no fundamental difference, since the heating released by convection is considered a source of Rossby waves, regardless of the source of the convection. However, ENSO can also influence teleconnections by modifying jet streams, whereas the MJO is unlikely to do so under the current climate. It is currently in the manuscript line 47-49.

L60: Remove "this" between "in" and "highly".

Thanks for your suggestion. We have removed the word 'this' in the revised manuscript.

L100: Replace "resolution" with "grid spacing" as the two are not synonymous (i.e., features are resolved at 4–6 times the grid spacing).

Thank you. We have replaced resolution to make the terms consistent.

L100–101: Given the time spent in the Introduction on diabatic heating influences, it would be worth a short discussion (perhaps in Section 4) on how this coarse grid spacing (and therefore, lack of diabatic processes) may influence your results.

Thanks for your suggestion. We have added more discussion about the impact of the coarse grid spacing in the discussion. It is currently in the manuscript line 107-109, although it seems that topography parameterization seems to have a larger impact in the literature.

L122–125: How was the spatial extent for the HEAT SST anomalies chosen?

To replicate the temperature contrast between Siberia and the western Pacific Ocean, we defined the SST anomaly region to span 60 degrees of longitude in the midlatitudes, mimicking the order of magnitude longitudinal extent of an ocean basin or continent. We have added one sentence in the revised manuscript to explain this. It is currently in the manuscript line 130-132.

L155, L162: Suggest putting two equations on separate lines.

Thanks for your suggestion. We have put two equations on separate lines in the revised manuscript.

L176: Should "CESM2" be "CAM6"?

Thanks for your suggestion. We have replaced CESM2 by CAM6 in the revised manuscript. It is currently in the manuscript line 187.

Fig. 1: I wonder if making each line slightly transparent would help the reader with interpretation, especially with identifying areas of overlap between experiments.

Thanks for your suggestion. That's a great suggestion. We have made the lines partially transparent to improve clarity when lines overlap. Figure 1 and Figure A1, A2 are both updated.

Fig. 2: Missing y-axis labels.

Thank you. This figure is removed based on the other reviewer's suggestion.

Fig. 2: Missing reference to panels (d), (e), and (f) in figure caption.

Thank you. This figure is removed based on the other reviewer's suggestion.

Fig. 3: Suggest including a pattern correlation coefficient analysis here to strengthen your ERA5- CNTL comparisons more objectively.

Thanks for your suggestion. We have added the pattern correlation coefficients to the upper right corner of the ERA5 figures in Figure 2 left column subplots.

L260: Specify which section "later" refers to.

Thanks for your suggestion. The "later" refers to subsections 3.2.1, 3.2.2 and 3.2.3 as a whole; it is currently in the manuscript line 248.

L279–281: Remove parenthetical sentence structure and split into two sentences to increase readability.

Thanks for your suggestion. We have removed the parenthetical content and use another sentence to illustrate it in the revised manuscript. It is currently in the manuscript line 248-251.

L316: Add "Figure" before "A7a to A7d".

Sorry for that and thanks for noting this. We have added 'Figure' wherever necessary in the revised manuscript. It is currently in the manuscript line 295.

L320: Suggest reordering Appendix figures so they are referenced in order in the text.

Thanks for your suggestion. We have double checked and reorder the Appendix figures in the revised manuscript.

L326: Should the last "value" be "Ks value"?

Thanks for your suggestion. We have specified the 'Ks value' explicitly at line 304, and we have checked that all Ks values are clearly defined in the revised manuscript.

Fig. 6: Suggest smoothing contours to help reduce noise and highlight signals.

Thanks for your suggestion. We have smoothed contours in the revised manuscript. Figure 5 and Figure A9 are both centered smoothed every 3 grid points.

L355: Is this really true for (a)? The correlation is only 0.19.

Thanks for pointing this out. We changed our argument a little bit, from arguing the importance of transient eddies for topography's impact to transient eddies is not so important in line 337-338. More discussion on the role of topography and SST patterns is added in the manuscript line 365-372.

L357–358: It appears the right correlations are weaker in (a) and (c) but right and left correlations in (b) and (d) are fairly similar. Please add additional discussion to explain this difference in result.

Thanks for your suggestion. We have discussed the potential distinct impacts of topography and diabatic heating in the revised manuscript, which also aligns with our planned response to a related comment from another reviewer. It is added in the manuscript line 365-372

L440: Acronym already defined in L70.

Thank you. We have removed the second reference to the acronym in the revised manuscript.

Section 5: The conclusions would benefit from bringing the presented results back to some of the implications of QSWs discussed in the Introduction. For example, what do these identified relationships between QSWs and certain variables mean for, say, forecasting QSWs and their effects on weather conditions?

Thanks for your suggestion. We certainly consider that aspect, but mostly about understanding the QSWs, and the bias of QSWs in climate models or forecast models. We've briefly mentioned a potential implication to understand the future QSWs distribution and its uncertainty at the end of conclusion.

**Reviewer 2:**

**General Comments:**

The paper presents an investigation into the Quasi Stationary Wave Strength and its dependence on surface forcing and internal dynamical factors. The investigation is performed using eight 100 year simulations from the CAM6 model. I believe the subject is of interest to the community and the set of simulations has been systematically/logically designed to probe the relationship between different surface forcings and Quasi Stationary Waves.

I found the paper quite difficult to read and spent a lot of time flicking back and forth between various sections to work out what was plotted in the various figures and why, so feel it would benefit from a careful revision to the text to make it more explicit in terms of analysis and interpretation and some restructuring to make the main scientific questions/hypotheses and methods clear. It is possible the paper is attempting to cover too much material so there are many multi-panel figures, which require a lot of explanation. Reducing the scope of the paper would in my view make it much clearer and stronger.

We thank the reviewer for taking the time to review our paper. Their comments and suggestions have significantly improved the manuscript. In response to this comment, we have removed the current Section 3.2 (on the role of zonal-mean zonal wind and meridional wind) to improve the readability and reduce the scope. We have also added more analysis to illustrate the distribution of quasi-stationary waves (QSWs) through temporal analysis, as suggested by the reviewer - the results from this analysis are consistent with the conclusions from the spatial correlation analysis. Additionally, we have outlined and

summarized the hypotheses, analysis, and reasoning/conclusions as bullet points to better highlight the core message of the paper.

The main body of the paper is devoted to testing two hypothesis:

- 1 The spatial distribution of quasi-stationary waves is governed by that of stationary waves.
- 2 The spatial distribution of quasi-stationary waves is governed by that of dynamic factors such as Eady growth rate and local stationary wavenumber and transient wave strength. These hypotheses appear in the results section, but I think they should really appear in the introduction motivated by/linked to the discussion of literature. The methods section can then more clearly explain how the methods are designed to address the hypotheses

Thanks for your suggestion. We fit the hypotheses in the introduction as suggested. In addition to these two main hypotheses on the distribution of QSWs, we also explore the impact of stationary forcings on the duration of QSWs, and keep this analysis in the manuscript, as we believe it's an important result for this work.

The partition between stationary, quasi-stationary and transient waves defined by the authors is as follows:

- Stationary waves: A day of year climatological mean.
- Quasi-stationary waves: 15 day low pass filtered anomalies the climatology
- Transient waves: 15 day high pass filtered anomalies to the climatology

All three of these are converted to wave-envelope amplitude (as in Zimin et al 2003) including only wavenumbers 4-15.

A major methodology employed in the paper is comparison of climatological mean spatial patterns of quasi-stationary wave amplitude to that of stationary waves and transient waves as well as to climatological mean spatial patterns of Eady growth rate and stationary wavenumber. So a number of the conclusions hang on the assumption that a similar climatological mean pattern implies a significant relationship between quantities/lack of a similar pattern implies no relationship. I think this method is employed in all figures except fig 1and fig 5 which makes use of time variation in the data. The robustness of this basic assumption requires some discussion in the text - it isn't clear from what is written why this is a useful method and what its possible limitations are. The authors could also consider including more results which more carefully examine the relationship between stationary wavenumber and Quasi-stationary wave strength by utilising information about time covariation of these two quantities - this to me would add weight to the argument that the two a closely linked.

Thank you for your important suggestion. We have added a little bit of discussion of this method and its limitations, especially in confirming the causality between QSWs and transient eddies, and new temporal analyses in discussion (4.2), including temporal correlations between QSW strength, stationary wavenumber Ks, and transient eddy strength. We have added the suggested temporal correlation analysis, finding correlations consistent with the results from the spatial correlations, although the pattern varies across regions and experiments.

The main conclusion of the paper is that stationary wavenumber has a clear association with QSW strength suggesting a simple barotropic interpretation. This is an interesting conclusion, but the more clarity is needed in the explanations of results to justify this.

Thank you for your suggestion. We have changed the wording in the next version of the manuscript. Our argument is based on the fact that the stationary wavenumber Ks is derived from the dispersion relation for barotropic Rossby waves. Therefore, even if some nonlinearity is allowed in the process, the QSWs should remain (equivalent) barotropic throughout their lifecycle. However, this only works for QSWs within midlatitudes. We have discussed more limitations in the last paragraph.

**Specific Comments:**

- There is a long introduction which covers well the literature in the area. This ends with a short paragraph about the scientific question/hypothesis the authors' work seeks to address. The hypothesis in this final paragraph is too vague and doesn't really explain what the authors aims are and how these fit into the literature described prior to this.

Thanks for your suggestion. We have emphasized more about how these stationary forcings affect circulation, and then may affect QSWs at the beginning of the last paragraph in the introduction (line 83-89) to explain the purpose of this work and why the previous literature is important. We have also specified the hypotheses at the end of introduction in bullet points as suggested in the next comment.

- Two hypotheses are given in section 3.3. The bulk of the paper seems to be aimed at testing these, so perhaps it would make sense to include these in the introduction and motivate them from the literature - why are these important and novel hypotheses to test. Then to include some discussion of the methods used to test these hypotheses in the methods section.

Thanks for your suggestion. We have stated the hypotheses and reasoning at the end of introduction.

- section 3.1 line 170 What does "frequency of all events is fixed" mean - explain this more clearly. Do you mean you are defining an amplitude threshold for an "event" such that the total number of events is the same between the different simulations and reanalysis?

Thanks for your suggestion. Indeed – in each dataset (reanalysis or simulation), we select the strongest 90 QSW events, corresponding to the 90 analyzed years, to ensure a fixed frequency of, on average, one QSW event per year. However, in response to other comments about reducing the panels in this figure, we have removed this analysis result from the main paper, and keep it only in the Appendix. This analysis shows that our results on duration are not sensitive to how we select events in the different simulations.

- Main conclusions of Figure 1 seem to be that the model is biased towards shorter duration events, including stronger forcing increases the duration of events. You could consider whether these points could be demonstrated with a simpler figure with fewer panels. It would also be good to comment on whether these conclusions would be expected from what we know about climate models/other modelling studies or are specific to this particular model.

Thank you for your suggestion. We have moved the fixed frequency results to the Appendix. We have also expanded the discussion on the performance of the CTRL experiment, including its limitations—such as the shortcomings of using prescribed SSTs, and connect these to known biases of climate models in the manuscript line 201-203.

- section 3.2 you state that amplitude of meridional wind is used as proxy for stationary wave amplitude. This implies you are calculating the zonal mean of magnitude of the time mean meridional wind, not the time-mean zonal-mean of the magnitude of the meridional wind which would include transient waves. However you don't state clearly in the text what is calculated.

Thanks for your suggestion. We've removed this section now.

- line 257: "As a first test, we calculate Spearman rank correlation coefficients". This appears to be the main test that is used in the paper to link QSWs to the other quantities.

Thank you for your comment. Our distinction was between the first test – correlation on the absolute fields from simulations, and the second test – correlation on the differences between experiments. We now also include the temporal analysis, as suggested by the reviewer, and so change the description in the manuscript line 236-238.

- Figure 3. Panels a and b are very clear, but it would be good to state the contour interval where they are described in the caption rather than at the end.

Thanks for your suggestion. We've changed the subplots in Figure 2 now.

- The effects of different types of surface on QSWs are investigated by examining differences in climatological means between different pairs of simulations, rather than just looking at anomalies from the control simulation. This makes the results quite difficult to

follow, my question is whether this complication is necessary or anomalies from the control could be used instead or whether the anomalies are necessary at all - i.e. could you make the same points just by looking at the fields themselves and comparing? This might simplify the analysis. If it is necessary to look at these anomalies as in the current paper to make the points, then it would be good to give more explanation to help the reader understand what they are learning from the different pairings.

Thank you for your suggestion. As mentioned above, we compare differences in climatological means between different pairs of simulations because (1) each metric shows some degree of correlation with QSW strength when comparing climatological values directly, but the differences do not necessarily (see Table 2), and (2) we aim to understand whether different stationary forcings\_(e.g. topography vs diabatic heating) influence QSWs in distinct ways. We have revised the description to clarify the reasoning behind this methodology (line 240-248), and on the potentially different impacts of diabatic heating and topography on QSWs (365-372, 472-480).

- A significant portion of the paper addresses the link between stationary wavenumber and quasi-stationary wave amplitude. This to me was the more interesting part (particularly fig 5), however the method for calculating stationary wave number is missing some key details: the authors state it is calculated from zonal wind, but don't specify whether this is a single level a vertical integral. In figures 3c and 3d, I can not determine what the contours are showing. At first I assumed from the caption/title that these were the stationary wavenumber with only values 4 to 8 shown, however the contours range from 2 to 10 so is this just stationary wavenumber?, if so why have 4-8 in the title - please provide a clear explanation.

Thanks for your suggestion. We calculate the stationary wavenumber Ks at 200 hPa, the same as the level where QSWs strength is calculated. We have added it in the revised Methods line 161. Thanks for pointing out the incorrect figure - we have corrected it in a revised manuscript Figure 2.

- Figure 5 is very interesting and appears to show a clear link between the Ks and the QSW amplitude. However it is not clear from the caption of the text what is actually plotted here. The count refers to the number of days in DJF which is clear and one assumes this sums up to 90xlength of simulation in years. But how do the authors decide what the QSW strength and stationary wavenumber are for a given day - is this a spatial mean value?

Thank you for your suggestion. The caption is incorrect—it should refer to the count of grid points, not days. The values of both stationary wavenumber Ks and QSW strength shown are climatological values. However, as per your earlier suggestion, we have added more temporal analyses. We've calculated the temporal correlation between Ks values and QSW strength in 3.2.3, Fig. 8 and correlations between transient eddies strength and QSWs in Fig. A10.

---

## Author Response (AR2)

I have copied the major comments of the reviewer in black, along with my responses in blue.

**Reviewer 1:**

Thank you to the authors for their work addressing comments raised by myself and another reviewer. The changes to the text and figures help the paper read more clearly and help better emphasize the study's main hypotheses and conclusions. I have just a couple of (very) minor suggestions to address prior to publication:

Thank you for your careful review and kind words. We appreciate your suggestions and will address the minor points you've noted prior to publication.

L337, L364: Should "transient eddies strength" be "transient eddy strength" or "strength of transient eddies"?

Thanks for pointing this out. We've changed all the "transient eddies strength" to "transient eddy strength" now.

L389: Can remove "FDR" acronym since it's not used elsewhere in the paper.

We've removed the abbreviation.

L392: Remove "shows" between "with" and "limited".

Thanks for pointing this out. We've removed the "shows" now.

Fig. 8: This figure is a bit hard to read as the geographical boundaries are largely masked by the stippling and contours. A couple of suggestions to try: (1) use dots for p<0.05 significance and omit p<0.1 stippling (or include in the appendix), (2) decrease weight of QSW strength contours.

Thank you for your suggestion. We have updated Fig. 8 and Fig. A10 to use dots for p<0.05 significance, omit stippling for p<0.1, and reduce the line weight of the QSW strength contours. Using dots rather than hatched lines is very helpful.

**Reviewer 2:**

This revised version of the paper has improved significantly the presentation of the results, making the authors aims and methods clear. The topic and results of the paper I think will be of interest to the community.

Thank you for your positive feedback. We appreciate your comments and are glad that the revisions have clarified our aims and methods.

A few minor corrections:

Title: temperatures should be temperature

Thanks for pointing it out. The plural form in title and abstract have been removed.

Equation 1 has no left-hand side.

Thanks for pointing this out. We've added the left-hand side and included a brief description below.